# Maternal anemia and underweight as determinants of pregnancy outcomes: cohort study in eastern rural Maharashtra, India

Archana Patel,[1] Amber Abhijeet Prakash,[1] Prabir Kumar Das,[1] Swarnim Gupta,[1] Yamini Vinod Pusdekar,[1] Patricia L Hibberd[2]

[1]Lata Medical Research Foundation, Nagpur, Maharashtra, India
[2]Boston University School of Public Health and Boston University School of Medicine, Boston, Massachusetts, USA

**Correspondence to**
Dr Amber Abhijeet Prakash; rockman.blues21@gmail.com

## ABSTRACT

**Objectives** To study the trend in the prevalence of anaemia and low BMI among pregnant women from Eastern Maharashtra and evaluate if low BMI and anaemia affect pregnancy outcomes.

**Design** Prospective observational cohort study.

**Setting** Catchment areas of 20 rural primary health centres in four eastern districts of Maharashtra State, India.

**Participants** 72 750 women from the Nagpur site of Maternal and Newborn Health Registry of NIH's Global Network, enrolled from 2009 to 2016.

**Main outcome measures** Mode of delivery, pregnancy related complications at delivery, stillbirths, neonatal deaths and low birth weight (LBW) in babies.

**Results** Over 90% of the women included in the study were anaemic and over a third were underweight (BMI <18 kg/m2) and with both conditions. Mild anaemia at any time during delivery significantly increased the risk (Risk ratio; 95% confidence interval (RR;(95% CI)) of stillbirth (1.3 (1.1–1.6)), neonatal deaths (1.3 (1–1.6)) and LBW babies (1.1 (1–1.2)). The risks became even more significant and increased further with moderate/severe anaemia any time during pregnancy for stillbirth (1.4 (1.2–1.8)), neonatal deaths (1.7 (1.3–2.1)) and LBW babies (1.3 (1.2–1.4)).,. Underweight at anytime during pregnancy increased the risk of neonatal deaths (1.1 (1–1.3)) and LBW babies (1.2;(1.2–1.3)).The risk of having stillbirths (1.5;(1.2–1.8)), neonatal deaths (1.7;(1.3–2.3)) and LBW babies (1.5;(1.4–1.6)) was highest when - the anaemia and underweight co-existed in the included women. Obesity/overweight during pregnancy increased the risk of maternal complications at delivery (1.6;(1.5–1.7)) and of caesarean section (1.5;(1.4–1.6)) and reduced the risk of LBW babies 0.8 (0.8–0.9)).

**Conclusion** Maternal anaemia is associated with enhanced risk of stillbirth, neonatal deaths and LBW. The risks increased if anaemia and underweight were present simultaneously.

**Trial registration number** NCT01073475.

## Strengths and limitations of this study

► This is a large prospective study with a sample size of 72 750 pregnant women collected over an 8-year period in the target geographical region of eastern Maharashtra, focusing on the synergistic effects of low body mass index (BMI) and anaemia anytime in pregnant women, on stillbirths, neonatal death and low birth weight in newborns, which may have been overlooked in prior research.

► Dietary practices and details of compliance with iron supplementation data were not collected in this study.

► Prepregnancy data on BMI and anaemia were not available in this study. The broad of range of gestational age during first visit mitigates the exclusive pathophysiological effect of undernutrition in women with low BMI due to undernutrition on pregnancy outcomes.

► The haemoglobin levels were measured using Sahli's method, which may not be accurate due to subjectivity of the method, poor sensitivity and lack of reliability.

restricts fetal growth, contributing to about 800 000 neonatal and 400 000 infant deaths, and 20% of stunting in the first 2 years of the child's life, as well as 20% of maternal deaths at delivery.[1–3] The Sustainable Development Goals bring explicit attention to nutrition, including the World Health Assembly target to reduce anaemia in women of ages 15–49 years by 50% by 2025.[4 5] Since India has largest number of neonatal, infant and under age 5 children deaths in the world as well as high rates of stunting and growth faltering,[6–8] it is particularly timely to understand maternal nutritional status as a risk factor for maternal and childhood adverse outcomes of pregnancy in the Indian population.

Two important indicators of maternal nutrition are body mass index (BMI) and anaemia, both of which can affect health of a mother

## INTRODUCTION

Malnutrition is a serious underlying cause for child and maternal deaths around the globe. Undernutrition during pregnancy

and her fetus. Much attention has been focused on complications of maternal overweight and obesity (BMI >25 kg/m$^2$) including gestational hypertension, pre-eclampsia, macrosomia, early induction of labour and need for caesarian deliveries and currently apply mostly to high but also recognised increasingly in middle-income countries, including India.[9] However, in rural India, undernutrition (BMI <18.5 kg/m$^2$) predominates and is associated with low birth weight (LBW; <2.5 kg) and preterm deliveries (<37 weeks of gestation). Prepregnancy BMI is most indicative of maternal undernutrition, but these data are rarely available when a woman enrols in the antenatal clinics in many developing countries including India, where the first antenatal visit is often after the first trimester.[10] Although the pathophysiology of low BMI early in pregnancy may differ from the same later in pregnancy (fetal growth faltering, oligohydramnios and so on), there is little information, if any, on the impact of low BMI recorded at the first antenatal visit regardless of GA on maternal and fetal outcomes.[11] This information would help to identify high-risk pregnant women at their first antenatal visit. Similarly, maternal anaemia, as a result of undernutrition, and infections coexists with low BMI, particularly in Indian women of reproductive age. In the 2015 National Family and Health Survey (NFHS), 23% of Indian women had a BMI less than 18.5 kg/m$^2$ and 53% were anaemic (haemoglobin (Hb) <11 gm/dL).[12] Even higher rates were reported in rural areas of the state of Maharashtra.[13] Maternal anaemia is also associated with postpartum haemorrhage (PPH), LBW, small for gestational age (SGA) babies and perinatal death.[14] However, there is minimal information on the consequences of the combination of low BMI and anaemia in pregnant women in rural Maharashtra and whether women with this dual burden should be targeted for specific interventions. We used data from an ongoing Maternal and Newborn Health Registry (MNHR) based in rural areas of Nagpur, Maharashtra, India, to: (1) describe the prevalence and trends of maternal anaemia and low BMI from 2009 to 2016; (2) describe the maternal demographic characteristics associated with different levels of maternal Hb and BMI from 2009 to 2016; (3) evaluate whether low BMI and anaemia were independent or synergistic risk factors for adverse pregnancy outcomes between 2009 and 2016; and (4) investigate whether low BMI and anaemia had additive or synergistic effects on poor birth outcomes.

## METHODS

### Study design and settings

The present study uses data collected from the Nagpur, India, site of MNHR, which is a population-based registry established in 2009 by the Global Network (GN) with support from *Eunice Kennedy Shriver* National Institute of Child Health and Human Development (NICHD), USA. The MNHR is an observational, prospective cohort operating in seven different sites around the world including two sites in India. The Nagpur site comprises of 20 study clusters in the four districts of Nagpur, Bhandara, Chandrapur and Wardha in eastern Maharashtra, India. Geographical areas surrounding government primary health centres (PHCs) that have approximately 900 deliveries per year constitute a study cluster. The MNHR aims to recruit all consenting women residing within the clusters, as early as possible during their pregnancy and then to follow them until 42 days postpartum. The MNHR methods have been published elsewhere.[15] The time period of this study was from June 2009 to December 2016.

### Patient and public involvement

The research questions for this study were developed to evaluate frequent reports of tiredness and weakness by the pregnant women enrolled in the prospective MNHR at the Nagpur site in rural eastern Maharashtra and to further review the reported high prevalence of maternal anaemia in the Indian Demographic Health Surveys. Specifically, we wanted to determine whether rates of anaemia had decreased over time, as a result of an active national iron supplementation programme, and whether in our prospective cohort, these rates had an impact on pregnancy outcomes. We also wanted to study whether low BMI, common in our study subjects, further exacerbated the impact of anaemia in pregnancy on perinatal outcomes. Since we intended to address these questions in a secondary data analysis using deidentified data in the MNHR, patients and public were not involved in the design of this study. The results will not be disseminated to study participants but are intended for additional research to find ways to reduce maternal undernutrition (low BMI) and anaemia, as well as to inform health policies on prepregnancy and maternal nutrition.

### Participants

The participants in this study were all consenting pregnant women residing in the 20 clusters of the Nagpur GN site, enrolled within the study period, who completed data collection (ie, had delivery outcomes and a completed 42 days postpartum follow-up) at the time of analyses. Women who had medical termination of pregnancy (MTP) (premature expulsion of a non-viable fetus before 20 weeks of gestation), miscarriages, multiple fetuses or had extreme/missing values for one or more study variables of interest were excluded from the analyses. Early enrolment of the participants was ensured through active surveillance in the study clusters by Nagpur site MNHR public health staff. The enrolled women were followed through to a perinatal visit after delivery (within a week) and to a follow-up visit around 42 days after birth. Maternal and infant data on study variables and outcomes were recorded for the enrollees.

### Data collection procedures

Briefly, data were collected by trained registry administrators (RAs) via standardised forms. RAs included medical officers and auxiliary nurse-midwives employed at PHCs

and subcentres. A 2-day training for data collection was provided to the RAs, followed by a refresher training every 6 months, and unscheduled special trainings as needed. The RAs returned filled in forms that were manually scrutinised for errors prior to data entry at Lata Medical Research Foundation, Nagpur. After entry, data were transmitted to the GN data management centre (Research Triangle Institute, Durham, North Carolina, USA) where digital data checks were performed on a monthly basis. The data management centre forwarded edit, monitoring and performance reports to the Nagpur site team for data cleaning and management. The data were continually reviewed for quality by NICHD appointed data monitoring committee and by the Nagpur site data collection team.

## Study variables

At the first antenatal visit, information regarding the enrolled women's characteristics including age, education, parity, height, weight and Hb was recorded. Weight to nearest 100 g using a spring balance and height to nearest centimetre was measured using a non-flexible measuring tape fixed to a wall. BMI was calculated by dividing weight in kilograms (kg) by squared height in metres. Standard WHO BMI classifications[16] were used to categorise the enrollees as underweight (BMI <18.5 kg/m$^2$), normal (BMI 18.5–24.9 kg/m$^2$) or as overweight/obese (BMI ≥25 kg/m$^2$). Hb was estimated within 2 weeks of enrolment using Sahli's method.[17] A concentration of Hb level <11 g/dL indicated presence of anaemia. Based on WHO classification,[18] Hb readings were used to define the severity of anaemia as normal (≥11 g/dL), mild (10-11 g/dL) and moderate/severe (<10 g/dL). BMI and anaemia were categorised as ordinal variables for assessing their rates in this study population in consistency with standard public health practices. A four-level combination variable for BMI–anaemia was defined as based on anaemia-hemoglobin <11 (g/dL) (anaemic) versus ≥11 (g/dL) (non-anaemic) and BMI <18.5 (kg/m$^2$) (underweight) versus ≥18.5 (kg/m$^2$) (normal or not underweight), creating 'anaemic-normal', 'anaemic-underweight', 'non anaemic-normal' and 'non anaemic-underweight'. Mother's age was recorded at the time of enrolment. Maternal education of up to four grades was considered 'primary', grades 5–10 as 'secondary' and more than grade 10 as 'university'. Parity was recorded as whole numbers zero or more, depending on if the woman had been pregnant before enrolment or in pregnancy had carried a fetus to 20 weeks of gestation or not.

## Study outcomes

The impact of three-level variables of anaemia (normal, mild and moderate/severe) and BMI (underweight, normal and overweight/obese) was assessed on rates of mode of delivery, any pregnancy-related complications at delivery and postdelivery birth outcomes. While we did not assess the determinants of maternal mortality as maternal mortality rates were low, we described the causes

of death with respect to the level of anaemia and BMI status. Mode of delivery was recorded as caesarean section (CS) or non-caesarean (vaginal and assisted vaginal). Any pregnancy-related complications at delivery were recorded as presence of one or more of the following; obstructed/prolonged labour, severe antepartum haemorrhage, severe PPH, hypertension, pre-eclampsia and oblique/abnormal lie. Birth outcomes such as stillbirth, neonatal deaths (deaths of neonates from birth to 28 days of life) and LBW rates were assessed. Stillbirth rates were calculated as number of stillbirths per 1000 total births, while neonatal death rates were calculated as number of neonatal deaths per 1000 live births. Birth weight was measured within 24 hours of birth by using either a pan spring or a pan digital weighing scale. LBW was defined as neonatal birth weight less than 2500 g.

## Statistical analyses

Data were analysed using the STATA V.13.1 statistical package. Study variables of interest and their combinations, with years of enrolment were cross-tabulated in Stata and were transformed to graphs either in Stata itself or in Microsoft Excel. Pearson's correlation coefficient was used to study correlation between Hb concentrations and BMI values and was tested for significance. A univariate multinomial logistic regression, adjusting for clustering was used to identify association between the WHO levels of anaemia and BMI categories. Significance of trends over the progression of years of enrolment in any of the study variables was determined using Cuzick's non-parametric test[19] for trend across ordered groups. The year-wise rates of the outcomes (CS, pregnancy-related complication and LBW) were determined for each level of anaemia and category of BMI. Stillbirths and neonatal death rates were calculated per 1000 total births and per 1000 live births, respectively. To assess the association of maternal demographic characteristics, anaemia and BMI with the study outcomes, the data were divided in two data sets, one which included participants who had their first antenatal visit ≤20 weeks of GA and the second set that included participants who had their first antenatal visit after 20 weeks of GA. Both data sets were analysed using multivariable regression analyses solved using generalised estimating equations, adjusting for clustering within the PHCs. For the first set of regression models, explanatory variables included demographic characteristics: mother's age, education level and parity, along with the three levels of anaemia and three categories of BMI, as independent variables, while dependent variables included all five study outcomes. The regressions were also performed using Hb and BMI as continuous variables and compared with the results when these variables were considered as ordinal variables. The interaction of Hb and BMI was also assessed. A second set of models used demographic characteristics along with the constructed anaemia–BMI variable ('anaemic-normal', 'anaemic-underweight', 'non

anaemic-normal' and 'non-anaemic-underweight') as explanatory variables for three postnatal outcomes: stillbirths, neonatal death and LBW. These models assessed the impact of anaemia or underweight either alone or in combination on the three postnatal outcomes, as it is relevant from the public health viewpoint to understand the risks of adverse outcomes when a combination of comorbidities such as anaemia and underweight coexist.

## RESULTS
### Participants
During the study period, 72 750 women had delivery outcomes, and complete follow-up through day 42 postpartum was completed. Women with invalid or missing data on BMI or Hb (n=764) and miscarriage/MTP/multiple gestation (n=4412) were excluded.

### Baseline maternal characteristics
Almost 91% of the included women were anaemic (65811/72750), over a third were underweight (25571/72750, 35.1%) and over a third were both anaemic and underweight (23867/72750, 32.8%). Anaemia was severe in less than 0.2% and moderate in nearly 48% women. The gestational age at the first antenatal visit was ≤20 weeks for 72%, 21–28 weeks for 15% and >28 weeks for 13% of the study population. The study population mostly aged 20–29 years (93%); about half were primiparous (48%) versus multiparous (52%). As shown in table 1, over time there was an increase in the proportion of pregnant women in the older age group - 29 years and over, and also in the proportion of those women who completed higher levels of education. For parity, there was a reduction in proportions of parity >2 in 2016. The levels of anaemia showed no significant changes over time. The proportion of women who were just underweight (28%–38%)%) (figure 1) and underweight with anaemia (25%–35%)%) (figure 2) increased over time. We found a highly significant correlation (r=0.2; p<0.001) between Hb and BMI. Multinomial logistic regression with normal Hb concentrations as reference, showed highly significant (p<0.001) associations between women who were underweight and those with moderate/severe anaemia (risk ratio (RR): 1.9, 95% CI 1.6 to 2.2) as well as those with mild anaemia (RR: 1.3, 95% CI 1.2 to 1.5).

### Maternal mortality
There were 29 maternal deaths in the study population, 20 of whom had severe anaemia. Moderate/severe anaemia was identified as a primary cause of death in 1 and as a comorbid condition in 19. Of the total 29 deaths, anaemia and underweight was observed in 8, anaemia alone in 19, underweight alone in 1 and overweight/obese in 2. Reasons for maternal deaths, were haemorrhage (n=9, 31%), pre-eclampsia/eclampsia (n=7, 24%), infection (n=6, 21%) and other/missing (n=7, 22%).

### Study outcomes by years of enrolment
The rates of stillbirths, neonatal deaths and LBW babies increased with increasing severity of anaemia, with lower BMI and, if anaemia and underweight coexisted as compared with either condition alone, consistently in all years of enrolment. Rates of LBW babies decreased in overweight/obese mothers. The rates of CS and pregnancy related complications increased with increasing BMI. Rates of CS increased over time for all levels of anaemia and BMI (table 2).

### Study outcomes: regression
The risks of CS and pregnancy-related complications during delivery were significantly higher in non-anaemic women versus anaemic women for both data sets. Mild anaemia recorded ≤20 weeks had no impact on the outcomes, but when recorded after 20 weeks, mild anaemia increased the risk of stillbirths and neonatal deaths. Moderate/severe anaemia increased the risk of neonatal deaths and LBW when recorded ≤20 weeks and when present after 20 weeks, it also increased the risk of stillbirths (table 3A and B). The rates of LBW were significantly higher in underweight women anytime during pregnancy (table 3A, B and C). The risk of CS as well as pregnancy-related complications were higher in women who were overweight/obese recorded anytime during pregnancy, while the risk of LBW was reduced in the same group when recorded after 20 weeks of pregnancy (table 3C). The risks of stillbirth, neonatal deaths and LBW were the highest when women were both anaemic and underweight (table 4).

Nulliparous women had a significantly higher risk for all five outcomes, higher mother's age (>29 years) was associated with stillbirth, neonatal deaths and LBW, while lower maternal education level (secondary or less) was associated with stillbirths, neonatal deaths and LBW. Women <20 years were at lower risk for CS. Women with lower levels of education were also significantly less likely to have a CS or pregnancy-related complication (table 3C).

## DISCUSSION
Maternal malnutrition continues to be a silent emergency particularly in low-income and middle-income countries (LMICs) like India. We attempted to understand through present study the trends in the prevalence of anaemia and low BMI over the 8 years and the less understood combination of low maternal BMI and anaemia with pregnancy-related outcomes among rural women from eastern Maharashtra. We found that more than 90% of the pregnant women were anaemic, 35% were underweight (BMI <18.5) and nearly a third had both conditions. It was alarming that the rates of anaemia remained unchanged over 8 years (2009–2016) and those of underweight increased. The majority of women had mild to moderate anaemia, with severe anaemia in less than 0.2%. While mild anaemia recorded before 20 weeks of GA did not increase the risk of stillbirths, neonatal deaths or LBW, it

**Table 1** Baseline characteristics of women in the study by year of enrolment

| Characteristics | Years | | | | | | | | |
| --- | --- | --- | --- | --- | --- | --- | --- | --- | --- |
| | 2009 N (%) 6433 (100) | 2010 N (%) 10447 (100) | 2011 N (%) 9626 (100) | 2012 N (%) 9905 (100) | 2013 N (%) 9496 (100) | 2014 N (%) 9454 (100) | 2015 N (%) 10108 (100) | 2016 N (%) 7281 (100) | Total N (%) 72750 (100) |
| **Mother's age (years)*** | | | | | | | | | |
| <20 | 131 (2) | 209 (2) | 181 (1.9) | 188 (1.9) | 190 (2) | 193 (2) | 209 (2.1) | 157 (2.2) | 1458 (2) |
| 20–29 | 6044 (94) | 9867 (94.4) | 9056 (94.1) | 9269 (93.6) | 8834 (93) | 8743 (92.5) | 9266 (91.7) | 6640 (91.2) | 67719 (93.1) |
| >29 | 258 (4) | 371 (3.6) | 389 (4) | 448 (4.5) | 472 (5) | 518 (5.5) | 633 (6.3) | 484 (6.6) | 3573 (4.9) |
| **Mother's education (grades)*** | | | | | | | | | |
| Primary or less (≤4) | 1476 (22.9) | 2281 (21.8) | 1846 (19.2) | 1927 (19.5) | 1882 (19.8) | 693 (7.3) | 633 (6.3) | 329 (4.5) | 11067 (15.2) |
| Secondary (5–10) | 3814 (59.3) | 6210 (59.4) | 5818 (60.4) | 5880 (59.4) | 5432 (57.2) | 4962 (52.5) | 5101 (50.5) | 3522 (48.4) | 40739 (56) |
| University (>10) | 1139 (17.7) | 1956 (18.7) | 1958 (20.3) | 2071 (20.9) | 2172 (22.9) | 3768 (39.9) | 4358 (43.1) | 3415 (46.9) | 20837 (28.6) |
| Missing | 4 (0.1) | 0 (0) | 4 (0) | 27 (0.3) | 10 (0.1) | 31 (0.3) | 16 (0.2) | 15 (0.2) | 107 (0.1) |
| **Parity*** | | | | | | | | | |
| Nulliparous | 2905 (45.2) | 5118 (49) | 4706 (48.9) | 4817 (48.6) | 4384 (46.2) | 4530 (47.9) | 4967 (49.1) | 3598 (49.4) | 35025 (48.1) |
| 1–2 | 3300 (51.3) | 5032 (48.2) | 4669 (48.5) | 4827 (48.7) | 4907 (51.7) | 4721 (49.9) | 4888 (48.4) | 3544 (48.7) | 35888 (49.3) |
| >2 | 228 (3.5) | 297 (2.8) | 249 (2.6) | 261 (2.6) | 204 (2.1) | 197 (2.1) | 234 (2.3) | 137 (1.9) | 1807 (2.5) |
| Missing | 0 (0) | 0 (0) | 2 (0) | 0 (0) | 1 (0) | 6 (0.1) | 19 (0.2) | 2 (0) | 30 (0) |
| **Anaemia (g/dL)** | | | | | | | | | |
| Moderate/severe (<10) | 2962 (46) | 5283 (50.6) | 4620 (48) | 4730 (47.8) | 4682 (49.3) | 4418 (46.7) | 5058 (50) | 3457 (47.5) | 35210 (48.4) |
| Mild (10<11) | 2743 (42.6) | 4341 (41.6) | 4033 (41.9) | 4337 (43.8) | 4006 (42.2) | 4152 (43.9) | 4041 (40) | 2948 (40.5) | 30601 (42.1) |
| Normal (≥11) | 728 (11.3) | 823 (7.9) | 973 (10.1) | 838 (8.5) | 808 (8.5) | 884 (9.4) | 1009 (10) | 876 (12) | 6939 (9.5) |
| **BMI (kg/m²)*** | | | | | | | | | |
| Underweight (<18.5) | 1784 (27.7) | 3537 (33.9) | 3391 (35.2) | 3450 (34.8) | 3411 (35.9) | 3424 (36.2) | 3786 (37.5) | 2788 (38.3) | 25571 (35.1) |
| Normal (18.5–24.9) | 4426 (68.8) | 6558 (62.8) | 5931 (61.6) | 6061 (61.2) | 5699 (60) | 5619 (59.4) | 5789 (57.3) | 4095 (56.2) | 44178 (60.7) |
| Overweight or obese (≥25) | 223 (3.5) | 352 (3.4) | 304 (3.2) | 394 (4) | 386 (4.1) | 411 (4.3) | 533 (5.3) | 398 (5.5) | 3001 (4.1) |
| **Anaemia (g/dL)–BMI (kg/m²) factors** | | | | | | | | | |
| Anaemic (<11)–normal (≥18.5) | 4070 (63.3) | 6256 (59.9) | 5522 (57.4) | 5823 (58.8) | 5463 (57.5) | 5384 (56.9) | 5569 (55.1) | 3857 (53) | 41944 (57.7) |
| Anaemic (<11)–underweight (<18.5) | 1635 (25.4) | 3368 (32.2) | 3131 (32.5) | 3244 (32.8) | 3225 (34) | 3186 (33.7) | 3530 (34.9) | 2548 (35) | 23867 (32.8) |
| Non-anaemic (≥11)–normal (≥18.5) | 579 (9) | 654 (6.3) | 713 (7.4) | 632 (6.4) | 622 (6.6) | 646 (6.8) | 753 (7.4) | 636 (8.7) | 5235 (7.2) |
| Non-anaemic (≥11)–underweight (<18.5) | 149 (2.3) | 169 (1.6) | 260 (2.7) | 206 (2.1) | 186 (2) | 238 (2.5) | 256 (2.5) | 240 (3.3) | 1704 (2.3) |
| BMI (kg/m²) (mean/SD/IQR) | 20.1/2.6/3.4 | 19.7/2.6/3.2 | 19.6/2.6/3.4 | 19.7/2.7/3.5 | 19.7/2.8/3.6 | 19.7/2.9/3.6 | 19.7/3/3.7 | 19.7/3/3.7 | 19.7/2.8/3.6 |
| Haemoglobin (g/dL) (mean/SD/IQR) | 9.8/0.9/1 | 9.7/0.9/1 | 9.7/0.8/1.1 | 9.8/0.8/1 | 9.9/0.8/0.8 | 9.9/0.8/0.8 | 9.9/0.8/1 | 9.9/0.9/1 | 9.8/0.8/1 |

*Highly significant (p<0.001) non-parametric trend over years of enrolment.
BMI, body mass index.

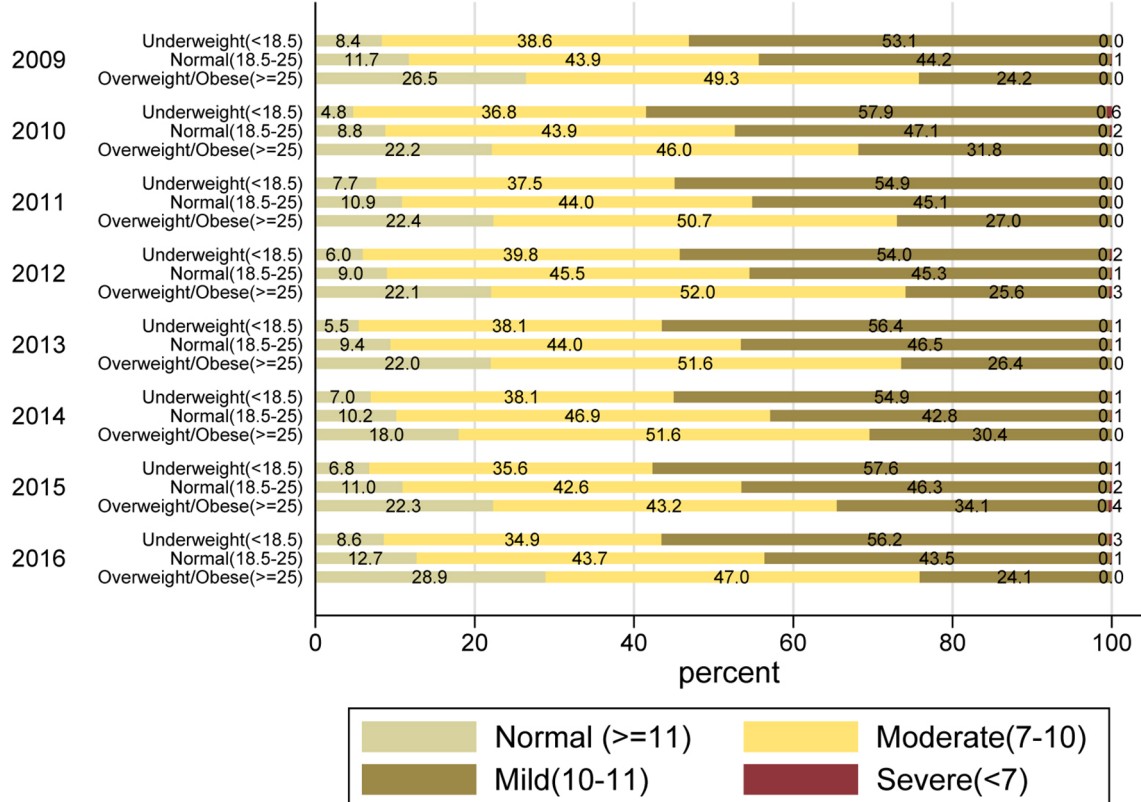

**Figure 1** Categorical anaemia (g/dL) in included women by body mass index (kg/m²) over year of enrolment.

increased the risk of stillbirths and neonatal deaths when recorded later in the pregnancy (>20 weeks). Moderate/severe anaemia recorded anytime during pregnancy increased the risk of neonatal deaths and LBW. It also increased the risk of stillbirths when recorded later in pregnancy. In developing countries, women often enrol late in the antenatal care services. In our study population, approximately a third enrolled after 20 weeks of pregnancy, 90% of which were anaemic and nearly half had moderate/severe anaemia.

Although anaemia occurring anytime during pregnancy is a risk factor for poor neonatal outcomes, anaemia especially during the third trimester is an important factor in determining birth weight. Rapid fetal growth occurs in

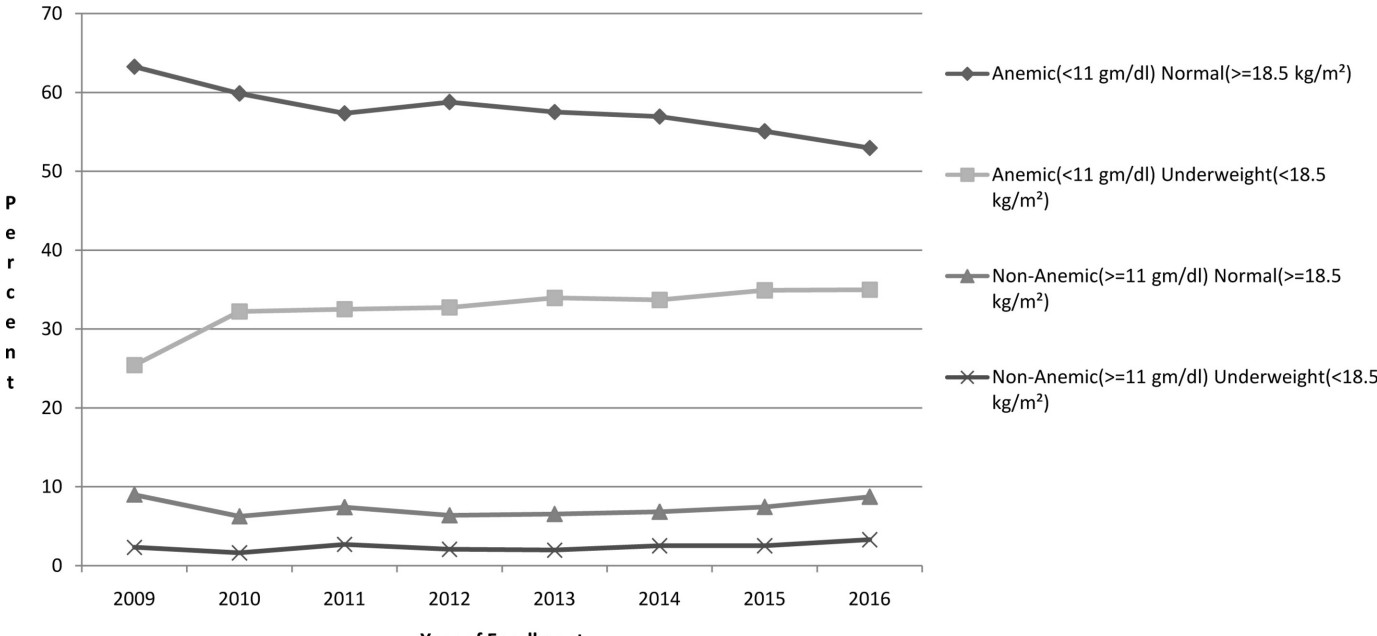

**Figure 2** Proportions of anaemia (g/dL) and body mass index (kg/m²) interaction in included women over year of enrolment.

**Table 2** Rates of study with respect to levels of anaemia and body mass index categories by year of enrolment

| Variables | Year of enrolment | | | | | | | | |
|---|---|---|---|---|---|---|---|---|---|
| | 2009 | 2010 | 2011 | 2012 | 2013 | 2014 | 2015 | 2016 | Total |
| **Rates of women experiencing caesarean sections** | | | | | | | | | |
| Anaemia (g/dL)* | | | | | | | | | |
| Moderate/severe (<10) | 15.67 | 16.83 | 17.19 | 19.53 | 20.91 | 23.74 | 24.73 | 28.67 | 20.85 |
| Mild (10–11) | 16 | 17.97 | 19.69 | 21.51 | 24.84 | 26.42 | 31.08 | 33.58 | 23.8 |
| Normal (≥11) | 18.41 | 26.12 | 24.87 | 26.13 | 31.44 | 32.01 | 37.07 | 39.61 | 29.8 |
| BMI (kg/m²) | | | | | | | | | |
| Underweight (<18.5) | 15.08 | 15.13 | 16.13 | 17.65 | 19.2 | 21.61 | 23.38 | 26.79 | 19.5 |
| Normal (18.5–24.9) | 15.95 | 18.65 | 19.91 | 21.53 | 24.64 | 26.87 | 29.9 | 33.41 | 23.6 |
| Overweight or obese (≥25) | 27.8 | 35.8 | 33.55 | 41.12 | 43.78 | 43.55 | 49.72 | 53.52 | 42.59 |
| Anaemic (HB <11 g/dL) and underweight (BMI <18.5 kg/m²)* | 15.05 | 14.96 | 15.94 | 17.32 | 18.98 | 21.19 | 23.31 | 26.14 | 19.22 |
| **Rates of women with pregnancy complications present at delivery** | | | | | | | | | |
| Anaemia (g/dL)* | | | | | | | | | |
| Moderate/severe (<10) | 14.58 | 14.1 | 12.03 | 14.76 | 14.72 | 10.5 | 10.7 | 10.85 | 12.78 |
| Mild (10<11) | 14.51 | 14.67 | 14.95 | 15.75 | 17.05 | 13.97 | 11.9 | 12.48 | 14.49 |
| Normal (≥11) | 16.62 | 21.51 | 16.65 | 17.42 | 22.15 | 18.1 | 15.86 | 14.27 | 17.73 |
| BMI (kg/m²) | | | | | | | | | |
| Underweight (<18.5) | 13.85 | 12.58 | 11.53 | 14.03 | 14.07 | 10.25 | 10.46 | 10.26 | 12.04 |
| Normal (18.5–24.9) | 14.91 | 15.46 | 14.37 | 15.38 | 17.07 | 13.54 | 11.99 | 12.43 | 14.48 |
| Overweight or obese (≥25) | 19.73 | 28.41 | 25.66 | 28.17 | 25.39 | 22.38 | 17.26 | 18.34 | 22.93 |
| Anaemic (HB <11 g/dL) and underweight (BMI <18.5 kg/m²)* | 13.82 | 12.44 | 11.85 | 14.09 | 13.64 | 9.79 | 10.28 | 10.09 | 11.92 |
| **Stillbirth rate per 1000 total births** | | | | | | | | | |
| Anaemia (g/dL) | | | | | | | | | |
| Moderate/severe (<10) | 29.37 | 25.55 | 22.29 | 23.89 | 26.27 | 21.05 | 23.33 | 26.61 | 24.54 |
| Mild (10–11) | 27.34 | 24.65 | 20.58 | 19.37 | 23.71 | 19.27 | 18.81 | 15.6 | 21.11 |
| Normal (≥11) | 12.36 | 18.23 | 15.42 | 16.71 | 16.09 | 16.97 | 14.87 | 15.98 | 15.85 |
| BMI (kg/m²)* | | | | | | | | | |
| Underweight (<18.5) | 29.15 | 25.73 | 22.41 | 22.32 | 25.21 | 20.74 | 22.45 | 27.62 | 24.05 |
| Normal (18.5–24.9) | 25.76 | 24.7 | 19.9 | 20.13 | 24.21 | 19.93 | 19.52 | 17.34 | 21.5 |
| Overweight or obese (≥25) | 22.42 | 11.36 | 23.03 | 30.46 | 18.13 | 12.17 | 20.64 | 10.05 | 18.33 |
| Anaemic (HB <11 g/dL) and underweight (BMI <18.5 kg/m²)* | 29.97 | 25.83 | 23.32 | 23.43 | 25.74 | 21.34 | 23.23 | 27.86 | 24.68 |
| **Neonatal death rate per 1000 live births** | | | | | | | | | |
| Variables | | | | | | | | | |

Continued

**Table 2** Continued

| | Year of enrolment | | | | | | | | |
| --- | --- | --- | --- | --- | --- | --- | --- | --- | --- |
| | 2009 | 2010 | 2011 | 2012 | 2013 | 2014 | 2015 | 2016 | Total |
| Anaemia (g/dL) | | | | | | | | | |
| Moderate/severe(<10) | 22.06 | 21.63 | 19.64 | 28.74 | 24.05 | 21.99 | 24.05 | 21.89 | 23.09 |
| Mild (10<11) | 17.54 | 16.08 | 15.44 | 16.74 | 19.03 | 18.26 | 17.72 | 16.47 | 17.15 |
| Normal (≥11) | 11.25 | 25.38 | 11.62 | 14.8 | 6.33 | 12.82 | 13.25 | 9.37 | 13.06 |
| BMI (kg/m²) | | | | | | | | | |
| Underweight (<18.5) | 21.23 | 20.13 | 21.26 | 23.68 | 24.98 | 20.09 | 22.94 | 20.74 | 21.96 |
| Normal (18.5–24.9) | 18.19 | 19.28 | 14.14 | 21.86 | 18.13 | 19.25 | 19.04 | 16.42 | 18.38 |
| Overweight or obese (≥25) | 13.95 | 20.53 | 27.68 | 15.96 | 13.37 | 17.54 | 17.54 | 18.13 | 17.97 |
| Anaemic (HB <11 gm/dL) and underweight (BMI <18.5 kg/m²) | 21.25 | 19.26 | 22.07 | 24.59 | 26.48 | 20.3 | 22.84 | 21.06 | 22.32 |
| **Variables** | Rates of low birth weight (<2500 g) babies | | | | | | | | |
| Anaemia (g/dL) | | | | | | | | | |
| Moderate/severe (<10) | 17.42 | 17.87 | 17.51 | 18.46 | 19.14 | 19.04 | 20.42 | 22.65 | 19.01 |
| Mild (10–11) | 16.99 | 14.37 | 14.63 | 13.53 | 14.5 | 14.72 | 16.98 | 18.69 | 15.35 |
| Normal (≥11) | 14.15 | 15.43 | 12.85 | 11.22 | 10.4 | 12.9 | 16.45 | 16.32 | 13.78 |
| BMI (kg/m²)* | | | | | | | | | |
| Underweight (<18.5) | 19.79 | 18.63 | 17.9 | 19.59 | 20.14 | 19.28 | 21.61 | 22.74 | 19.92 |
| Normal (18.5–24.9) | 15.95 | 15.23 | 14.69 | 13.81 | 14.6 | 15.09 | 17.29 | 19.54 | 15.61 |
| Overweight or obese (≥25) | 11.66 | 10.51 | 15.13 | 10.41 | 10.88 | 14.11 | 12.38 | 10.8 | 11.96 |
| Anaemic (HB <11 gm/dL) and underweight (BMI <18.5 kg/m²)* | 20.18 | 18.53 | 17.89 | 19.79 | 20.5 | 19.55 | 21.5 | 22.96 | 20.04 |

*Significant (p<0.01) non-parametric trend over years of enrolment.
BMI, body mass index; Hb, haemoglobin.

**Table 3A** Risk factors for study outcomes, multivariable estimates (GA at enrolment ≤20 weeks)

| Characteristics | Study outcomes | | | | |
|---|---|---|---|---|---|
| | Caesarean sections n=11 628, 24% RR (95% CI)† | Pregnancy-related maternal complications n=6762, 14% RR (95% CI)† | Stillbirths n=1104, 2.3% RR (95% CI)† | Neonatal deaths (within 28 days of birth) n=963, 2.0% RR (95% CI)† | Low birth weight (<2500 g) n=8378, 18% RR (95% CI)† |
| **Mother's age (years)** | | | | | |
| <20 | 0.83†* (0.73 to 0.95) | 0.83* (0.72 to to 0.97) | 0.73 (0.47 to 1.12) | 1.23 (0.87 to 1.75) | 1 (0.88 to –1.15) |
| 20–29 | Ref | Ref | Ref | Ref | Ref |
| >29 | 1.56** (1.44 to 1.68) | 1.37** (1.22 to 1.53) | 1.45* (1.13–1.86) | 1.37* (1.03 to 1.80) | 1.27** (1.15 to 1.40) |
| **Mother's education (grades)** | | | | | |
| Primary or less (≤4) | 0.50** (0.46 to 0.54) | 0.76** (0.70 to 0.83) | 1.68** (1.39 to 2.04) | 1.53** (1.24 to 1.87) | 1.17** (1.08 to 1.25) |
| Secondary (5–10) | 0.70** (0.67 to 0.73) | 0.90** (0.85 to 0.94) | 1.40** (1.21 to 1.61) | 1.30** (1.12 to 1.52) | 1.15** (1.09 to 1.21) |
| University (>10) | Ref | Ref | Ref | Ref | Ref |
| **Parity** | | | | | |
| Nulliparous | 1.33** (1.28 to 1.39) | 2.00** (1.86 to 2.15) | 1.32** (1.17 to 1.50) | 1.57** (1.38 to 1.80) | 1.38** (1.31 to 1.44) |
| 1–2 | Ref | Ref | Ref | Ref | Ref |
| >2 | 0.49** (0.40 to 0.60) | 0.80* (0.63 to 1.00) | 1.50* (1.06 to 2.13) | 1.62* (1.10 to 2.37) | 0.99 (0.85 to 1.17) |
| **Anaemia (g/dL)** | | | | | |
| Severe/moderate (<10) | 0.89** (0.84 to 0.95) | 0.90* (0.83 to 0.98) | 1.16 (0.92 to 1.46) | 1.37* (1.06 to 1.77) | 1.24** (1.14 to 1.35) |
| Mild (10–11) | 0.91* (0.86 to 0.97) | 0.92* (0.85 to 0.99) | 1.12 (0.88 to 1.41) | 1.16 (0.90 to 1.50) | 1.09 (1.00 to 1.18) |
| Normal (≥11) | Ref | Ref | Ref | Ref | Ref |
| **Body mass index (kg/m²)** | | | | | |
| Underweight (<18.5) | 0.81** (0.77 to 0.84) | 0.83** (0.79 to 0.88) | 1.08 (0.96 to 1.22) | 1.09 (0.96 to 1.24) | 1.18** (1.13 to 1.23) |
| Normal (18.5–24.9) | Ref | Ref | Ref | Ref | Ref |
| Overweight or obese (≥25) | 1.64** (1.51 to 1.78) | 1.44** (1.28 to 1.61) | 1.19 (0.84 to 1.67) | 1.25 (0.87 to 1.79) | 0.91 (0.79 to 1.04) |

*P<0.05, **p<0.001.
†Obtained by multivariable GEE regression models for each outcome, separately.
GA, gestational age; GEE, generalised estimating equations.

**Table 3B** Risk factors for study outcomes, multivariable estimates (GA at enrolment >20 weeks)

| Characteristics | Study outcomes | | | | |
| --- | --- | --- | --- | --- | --- |
| | Caesarean sections n=4816, 20% RR (95% CI)† | Pregnancy-related maternal complications n=3260, 14% RR (95% CI)† | Stillbirths n=488, 2% RR (95% CI)† | Neonatal deaths (within 28 days of birth) n=390, 1% RR (95% CI)† | Low birth weight (<2500 g) n=3776, 16% RR (95% CI)† |
| Mother's age (years) | | | | | |
| <20 | 0.63 *(0.46–0.86) | 0.78 (0.58 to 1.07) | 1.28 (0.66 to 2.49) | 1.47 (0.75 to 2.87) | 1.17 (0.92 to 1.48) |
| 20–29 | Ref | Ref | Ref | Ref | Ref |
| >29 | 1.66** (1.48 to 1.86) | 1.37** (1.18 to 1.60) | 2.31** (1.70 to 3.13) | 1.47 (1.00 to 2.16) | 1.29** (1.13 to 1.48) |
| Mother's education (grades) | | | | | |
| Primary or less (≤4) | 0.45** (0.40 to 0.50) | 0.61** (0.54 to 0.68) | 1.51* (1.14 to 2.01) | 1.56* (1.13 to 2.14) | 1.08 (0.98 to 1.19) |
| Secondary (5–10) | 0.66** (0.62 to 0.71) | 0.87** (0.80 to 0.94) | 1.34* (1.05 to 1.71) | 1.37* (1.04 to 1.80) | 1.10* (1.02 to 1.19) |
| University (>10) | Ref | Ref | Ref | Ref | Ref |
| Parity | | | | | |
| Nulliparous | 1.51** (1.41 to 1.60) | 1.98** (1.83 to 2.15) | 1.25* (1.04 to 1.51) | 1.31* (1.06 to 1.61) | 1.31** (1.23 to 1.40) |
| 1–2 | Ref | Ref | Ref | Ref | Ref |
| >2 | 0.42** (0.32 to 0.55) | 0.74* (0.56 to 0.98) | 0.89 (0.56 to 1.42) | 1.49 (0.95 to 2.34) | 1 (0.84 to 1.20) |
| Anaemia (g/dL) | | | | | |
| Severe/moderate (<10) | 0.79** (0.72 to 0.87) | 0.87* (0.78–0.97) | 2.35** (1.56 to 3.53) | 2.76** (1.73 to 4.42) | 1.33** (1.19 to 1.49) |
| Mild (10–11) | 0.87** (0.80 to 0.94) | 0.87* (0.78 to 0.97) | 1.81* (1.20 to 2.72) | 1.73* (1.08 to 2.76) | 1.07 (0.96 to 1.20) |
| Normal (≥11) | Ref | Ref | Ref | Ref | Ref |
| Body mass index (kg/m²) | | | | | |
| Underweight (<18.5) | 0.84** (0.77 to 0.91) | 0.82* (0.74 to 0.90) | 1.12 (0.91 to 1.39) | 1.12 (0.88 to 1.42) | 1.28** (1.19 to 1.38) |
| Normal (18.5 to 24.9) | Ref | Ref | Ref | Ref | Ref |
| Overweight or obese (≥25) | 1.59** (1.45 to 1.74) | 1.55** (1.38 to 1.73) | 0.7 (0.44 to 1.10) | 1.03 (0.67 to 1.60) | 0.78* (0.68 to 0.91) |

*P<0.05, **p<0.001.
†Obtained by multivariable GEE regression models for each outcome, separately.
GA, gestational age; GEE, generalised estimating equations.

**Table 3C** Risk factors for study outcomes, multivariable estimates (all included women)

| Characteristics | Study outcomes | | | | |
|---|---|---|---|---|---|
| | Caesarean sections n=16693, 23% RR (95% CI)† | Pregnancy-related maternal complications n=10163, 14% RR (95% CI)† | Stillbirths n=1620, 2.2% RR (95% CI)† | Neonatal deaths (within 28 days of birth) n=1368, 1.9% RR (95% CI)† | Low birth weight (<2500 g) n=12347, 17% RR (95% CI)† |
| Mother's age (years) | | | | | |
| <20 | 0.80** (0.72 to 0.90) | 0.83* (0.73 to 0.95) | 0.88 (0.62 to 1.25) | 1.28 (0.94 to 1.75) | 1.04 (0.92 to 1.16) |
| 20–29 | Ref | Ref | Ref | Ref | Ref |
| >29 | 1.60** (1.50 to 1.70) | 1.38** (1.26 to 1.50) | 1.76* (1.45 to 2.12) | 1.47** (1.18 to 1.83) | 1.28** (1.18 to 1.38) |
| Mother's education (grades) | | | | | |
| Primary or less (≤4) | 0.47** (0.44 to 0.50) | 0.69** (0.64 to 0.74) | 1.62** (1.38 to 1.89) | 1.49** (1.26 to 1.77) | 1.14** (1.08 to 1.21) |
| Secondary (5–10) | 0.68** (0.66 to 0.70) | 0.89** (0.85 to 0.93) | 1.38** (1.22 to 1.57) | 1.30** (1.14 to 1.49) | 1.14** (1.09 to 1.19) |
| University (>10) | Ref | Ref | Ref | Ref | Ref |
| Parity | | | | | |
| Nulliparous | 1.38** (1.34 to 1.43) | 2.00** (1.88 to 2.12) | 1.31** (1.18 to −1.45) | 1.48** (1.33 to 1.66) | 1.35** (1.30 to 1.41) |
| 1–2 | Ref | Ref | Ref | Ref | Ref |
| >2 | 0.45** (0.38 to 0.53) | 0.76* (0.64 to 0.90) | 1.22 (0.93 to 1.62) | 1.52* (1.14 to 2.04) | 1 (0.89 to 1.13) |
| Anaemia (g/dL) | | | | | |
| Severe/moderate (<10) | 0.87** (0.83 to 0.92) | 0.89** (0.84 to 0.95) | 1.43** (1.17 to 1.75) | 1.67** (1.33 to 2.08) | 1.26** (1.18 to 1.35) |
| Mild (10>11) | 0.90** (0.86 to 0.94) | 0.90* (0.85 to 0.96) | 1.28* (1.05 to 1.57) | 1.28* (1.02 to 1.60) | 1.08* (1.01 to 1.15) |
| Normal (≥11) | Ref | Ref | Ref | Ref | Ref |
| Body mass index (kg/m²) | | | | | |
| Underweight (<18.5) | 0.85** (0.82 to 0.88) | 0.85** (0.81 to 0.88) | 1.1 (1.00 to 1.22) | 1.12* (1.00 to 1.25) | 1.21** (1.16 to 1.25) |
| Normal (18.5 to 24.9) | Ref | Ref | Ref | Ref | rRef |
| Overweight or obese (≥25) | 1.57** (1.48 to 1.67) | 1.48** (1.36 to 1.60) | 0.92 (0.70 to 1.20) | 1.11 (0.84 to 1.46) | 0.84* (0.76 to 0.94) |

*P<0.05, **p<0.001.

†Obtained by multivariable GEE regression models for each outcome, separately.

GEE, generalised estimating equations.

**Table 4** Anemia–BMI factors for study outcomes, multivariable estimates

| Characteristics | Study outcomes | | |
| --- | --- | --- | --- |
| | Stillbirths n=1620, 2.2% RR (95% CI)† | Neonatal deaths (within 28 days of birth) n=1368, 1.9% RR (95% CI)† | Low birth weight (<2500 g) n=12 347, 17% RR (95% CI)† |
| Mother's age (years) | | | |
| <20 | 0.88 (0.62 to 1.25) | 1.28 (0.94 to 1.75) | 1.04 (0.93 to 1.16) |
| 20–29 | Ref | Ref | Ref |
| >29 | 1.75** (1.45 to 2.11) | 1.48** (1.19 to 1.84) | 1.27** (1.17 to 1.37) |
| Mother's education (grades) | | | |
| Primary or less (≤4) | 1.64** (1.40 to 1.91) | 1.52** (1.29 to 1.80) | 1.16** (1.10 to 1.23) |
| Secondary (5–10) | 1.40** (1.23 to 1.58) | 1.32** (1.16 to 1.51) | 1.15** (1.10 to 1.20) |
| University (>10) | Ref | Ref | Ref |
| Parity | | | |
| Nulliparous | 1.30** (1.18 to 1.44) | 1.47** (1.31 to 1.64) | 1.35** (1.29 to 1.40) |
| 1–2 | Ref | Ref | Ref |
| >2 | 1.24 (0.94 to 1.63) | 1.56* (1.17 to 2.09) | 1.02 (0.91 to 1.15) |
| Anaemia (g/dL)–BMI (kg/m$^2$) factors | | | |
| Anaemic (<11)–normal (≥18.5) | 1.30* (1.04 to 1.63) | 1.54** (1.19 to 2.00) | 1.22** (1.13 to 1.32) |
| Anaemic (<11)–underweight (<18.5) | 1.47** (1.17 to 1.85) | 1.74** (1.33 to 2.26) | 1.49** (1.37 to 1.62) |
| Non-Anaemic (≥11)–normal (≥18.5) | Ref | Ref | Ref |
| Non-Anaemic (≥11)–underweight (<18.5) | 0.96 (0.62 to 1.48) | 1.37 (0.88 to 2.13) | 1.37** (1.20 to 1.57) |

*P<0.05, **p<0.001.

†Obtained by univariate GEE regression models for each outcome, separately.

GEE, generalised estimating equations.

the third trimester, increasing the iron and other micronutrient requirement. This pathophysiology explains the association of third trimester Hb levels with LBW and neonatal deaths.[20 21]

This study shows that a subtle public health action of early and aggressive management of anaemia could lower the risk of adverse neonatal outcomes. Rates of underweight increased by 10% over 8 years, and there was an increase in the rates of overweight/obese from 2012 to 2016. Underweight recorded any time during pregnancy increased the risk of LBW (table 3A and B). A third of this study population was underweight any time during pregnancy, which is an underestimation of prepregnancy BMI. More than a third of women were both anaemic and underweight. The risk of stillbirths, neonatal deaths and LBW increased further if both anaemia and underweight coexisted (table 4). A limiting factor in our study was that we were not able to evaluate likely confounders of adverse outcomes in pregnancy such as socioeconomic status, number and timing of antenatal visits and use of tobacco and others as these data were not collected in the MNHR.

The NFHS data (NFHS-3 and NFHS-4) found anaemia prevalence rate of 54% between 2005 and 2016, with a minimal reduction in Maharashtra between NFHS-3 (58%) to NFHS-4 (49%).[22] Disturbingly, our estimates were higher than the District Level Household Survey (DLHS-4, 2012–2013), which reported a rate between 43% and 49% among pregnant women aged 15–49 years in the Nagpur, Bhandara and Wardha districts, but also remained static over the past 8 years[23–25] Previous studies from eastern Maharashtra have reported similar high rates of anaemia.[26–28] The women in our study were young (20–29 years), from both rural and backward tribal regions known for high prevalence of micronutrient deficiency specifically iron deficiency anaemia and other causes of anaemia such as sickle cell anaemia, hookworm infestation and malaria.[29–31] Variations in rates of anaemia may be due to methods used for surveying and assessing Hb levels.[32] The unchanging status of anaemia could be as a result of irregular and poor coverage of the national iron supplementation programme[33–35] and suboptimal compliance with iron supplementation during pregnancy (<30%),[10 36] due to side effects, forgetfulness to take timely dosage of iron supplementation pills and false beliefs about the importance of anaemia in pregnancy.[37–39] Directly observed administration of iron supplementation may improve compliance and is being implemented for adolescent girls in India.[40 41] We believe it should be considered in pregnant women. Some reduction in rates of severe anaemia were observed, which may be attributable to improved attendance of ante-natal care visits, early detection and

current management guidelines using injectable iron for treatment of severe anaemia at the public health facilities from the 12th 5-year plan.[42]

Our study reports over a third of the study participants to be underweight during pregnancy, which is consistent with the findings of NFHS for rural Nagpur.[13] We also found an 8% increase in low BMI in pregnant women between 2009 and 2016. There was also a trend towards BMI from the normal to overweight/obese categories over this same time period as has been observed.[43] Maternal prepregnancy BMI as well as BMI during any trimester of pregnancy has different effects on maternal and fetal outcomes. Adverse effects of low BMI preferentially target the fetus by causing growth restriction and LBW, whereas high BMI has a greater impact on maternal health and subsequently affects neonatal health through altered glucose homeostasis, leading to fetal macrosomia and other such pathophysiological mechanisms.[44–46]

Anaemia and underweight coexisted among one-third of the study participants, which increased over time. This change may be due to increasing prevalence of underweight as the rates of anaemia did not change over the study period. The most probable cause for maternal underweight in this region would be malnutrition.[47 48] It is also likely that iron and other micronutrient deficiency were responsible for higher rates of anaemia in underweight women,[49–51] but this should be studied further to reduce rates of anaemia in the community.

The rates of CS increased over the years consistent with an increase in facility-based deliveries in India.[52] However, anaemia at any level reduced the likelihood of CS and pregnancy-related complication, and underweight and anaemic women are more likely to have fetal growth retardation and smaller babies.[53–56] Pregnancy-related complication in our study population are most commonly due to obstructed labour and toxaemia of pregnancy, both of which are less likely to occur in underweight and anaemic women who tend to have smaller babies.[49] Women with higher BMI (who are less likely to be anaemic) are at increased risk of miscarriage, gestational diabetes, pre-eclampsia, venous thromboembolism, induced labour, CS, anaesthetic complications and wound infections.[48]

The risk of LBW was mostly determined by low BMI and severity of anaemia any time during pregnancy. The rates of LBW increased over the years in women who had moderate/severe anaemia and in women who were underweight. Other Indian studies have reported similar results.[9] The risk of having LBW babies in women who were underweight also increased by 20%, consistent with other studies that have reported an association of maternal anthropometry and size of the baby.[2 3] A study in Tanzania, on 12 269 newborns, identified short maternal stature as an significant risk factor for SGA and preterm appropriate for gestational age.[57] A recent study from Mexico reported that higher proportions of SGA neonates were born to malnourished and anaemic mothers than the mothers who were either malnourished or anaemic.[58] LBW babies are at increased risk of becoming stunted

adults, which will expound the intergenerational burden of LBW.[59]

The risk of stillbirths increased when any anaemia was recorded after 20 weeks of GA. While there are studies to support our findings on the association of maternal anaemia with stillbirths,[12 60 61] it is unclear whether it was a result of an intrapartum complication or whether it reflected poor socioeconomic conditions that limited access to quality intrapartum care. It is also possible that sickle cell anaemia, hookworm infestation and malaria, all common in eastern Maharashtra, may have contributed to higher rates of stillbirths,[22–25] as does ethnicity in underweight women previously.[62]

The risk of neonatal deaths increased when mild anaemia was recorded after 20 weeks of GA and if moderate/severe anaemia was recorded at any time during pregnancy. Although there is a paucity of data on the reason for an association between anaemia and neonatal mortality, it is consistent with the finding of a recent meta-analysis in LMIC (one study from India and another from Malawi) that found the odds of neonatal mortality to be 2.7 times higher among the anaemic mothers.[55 59 63] The neonatal mortality could also be attributed to increased risk of preterm and LBW deliveries in women who are anaemic.[64] Underweight recorded any time during pregnancy did not increase neonatal death. However, when the two conditions coexisted, the risk was higher than presence of either one of them alone (table 4). Low maternal BMI and anaemia are associated with LBW and preterm deliveries, which are independent risk factors for infant mortality in India.[12 65 66]

## CONCLUSIONS AND PUBLIC HEALTH IMPLICATIONS

The combination of anaemia and underweight in pregnancy increased the risk of stillbirths, neonatal deaths and delivering LBW babies. These findings raise major concerns because national programmes to address iron deficiency anaemia have not reduced the rates of anaemia among the rural pregnant women in eastern Maharashtra over past 8 years. There has been a steady increase in prevalence of underweight in pregnancy with almost a third of pregnant women being both anaemic and underweight. Immediate and effective public health interventions are needed to address maternal anaemia and malnutrition to improve birth outcomes. There is an urgent need for evaluation and addressing of gaps of the iron supplementation programme for adolescent girls and pregnant women in these areas with high rates of anaemia, in addition to a more holistic approach to curb maternal malnutrition. Meanwhile, seeking simultaneously short-term strategies such as referral to a better healthcare service and management of delivery for the segment of pregnant population who are both anaemic and underweight could be helpful in reducing neonatal mortality. Such actions are imperative to break the intergenerational cycle of poor growth in the offspring and also for improving child survival.

**Acknowledgements** We are deeply indebted to Research Triangle International, Durham, USA, for their enthusiastic support. Thanks and recognition are due to the Registry Administrators at the Nagpur site of the Maternal and Newborn Health Registry (MNHR) who collect data, often in very challenging circumstances, and the communities, families and mothers who agree to participate in the MNHR. We express our sincere gratitude to Dr Vaishali Khedikar, Dr Savita Bhargav and Dr Kunal Kurhe at Lata Medical Research Foundation, Nagpur, who helped immensely by strengthening the manuscript through their inputs.

**Contributors** AP conceived and designed the study and developed the initial data collection tools; AP and AAP wrote the first draft of the manuscript, which PLH, PKD, YVP and SG subsequently revised. AP and AAP developed the analysis plan outline and AAP carried out the data analyses. AP, PLH, PKD, YVP and SG contributed to data analysis through review, interpretation and cross-checking figures and tables. All authors read, revised, and approved the final manuscript.

**Funding** This study was funded by the *Eunice Kennedy Shriver* National Institute of Child Health and Human Development of the US National Institutes of Health. (Grant Reference Nos. U01HD058322 and U01HD078439).

**Competing interests** None declared.

**Patient consent** Not required.

**Ethics approval** The study in Nagpur, India, has been reviewed and approved by the Institutional Review Board of the Lata Medical Research Foundation (RPC#22A) and the Partners Human Research Committee/IRB in Boston, Massachusetts (H-35430).

**Provenance and peer review** Not commissioned; externally peer reviewed.

**Data sharing statement** A deidentified minimal dataset that underlies the findings and conclusions described in the manuscript can be shared upon request by the editors to verify the reported study findings. All authors hereby declare that individual participant data in any form is not a part of this publication.

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
