## [Reviewer comments · BMJ Open]

ARTICLE DETAILS

TITLE (PROVISIONAL)	MATERNAL ANEMIA AND UNDERWEIGHT AS DETERMINANTS OF PREGNANCY OUTCOMES: COHORT STUDY IN EASTERN RURAL MAHARASHTRA, INDIA
AUTHORS	Patel, Archana; Prakash, Amber; Das, Prabir Kumar; Gupta, Swarnim; Pusdekar, Yamini; Hibberd, Patricia

VERSION 1 – REVIEW

REVIEWER	Dr. Prakash Prabhakarrrao Doke Department of Community Medicine, Bharati Vidyapeeth Deemed University, Medical College, Pune, India
REVIEW RETURNED	23-Jan-2018

GENERAL COMMENTS	Title mentions 'rural Maharashtra'; the article pertains to only four districts from Nagpur division. Title mentions pregnancy outcomes; however the article does not include abortion which is very important and adverse outcome of pregnancy. The title says it is a cohort study. In cohort study readers expect analysis of relative risk. The relative risk is mentioned only in abstract and risk ratio at some places in results. Attributable and population attributable risk was expected. The study description does not satisfy the expectations. At the end the check list for STROBE statement for cross sectional studies is given instead of cohort study. There should be some uniformity. The objective in abstract mentions 'Central India'. It seems inconsistent. Mention of mode of delivery as the first main outcome does not get appropriate attention in subsequent writing. If low BMI and anemia are independent variables, mode of delivery may be distantly associated. For mode of delivery availability facilities for cesarean sections is the determining variable.
---

	The result section gives relative risks associated with anemia and underweight separately but risks are not given when they are coexisting. In limitations some mention is there of pre-pregnancy status. In introduction SDGs and Global Nutrition Targets are mentioned. Addition of strategies in Indian context will be more relevant. In twelfth plan reduction in anemia in pregnancy is targeted. In objectives, the word measure prevalence of anemia may be more suited because reasons, laboratory investigations etc. are not part of study. In second objective the words levels of maternal hemoglobin are used but excepting categorization of anemia no quantitative correlation is attempted. Either the words should be replaced by anemia or depending upon of level hemoglobin serial relative risks may be provided. Study settings may mention named of district. District wise information both from DLHS-4 and NFHS-4 is available. The findings from these reports need to be considered for better comparison. The criteria of excluding termination of pregnancy may underestimate the risk of adverse pregnancy outcome associated with anemia/underweight. The women who were followed upto 42 days were eligible for inclusion. Limitations of Sahli's method of estimation of hemoglobin may be have been mentioned. The WHO reference quoted for classification of anemia based up on hemoglobin level is very old may be replaced by latest one. Analysis by three categories of anemia universally accepted may give more meaningful information about graded response. Secondly the national guidelines for management anemia vary as per severity. The five study outcomes mentioned are not in consonance with objectives. Mode of delivery, pregnancy related complications and neonatal death do not fit aptly under pregnancy outcomes. Instead of classifying mode of delivery into cesarean and non-cesarean, cesarean and vaginal may have been more appropriate terms. In result section one of the reasons given for exclusion is no pregnancy outcome yet. This is contradictory to inclusion criteria. These women did not meet criteria of inclusion. Probably these were
--	--

	the pregnant women enrolled in last few months. They need not be included in first instance. A slight but significant correlation was observed which may be due to large number. The observation of more probability of maternal complications and cesarean rate among non-anemic women and less in anemic is contradictory to prevalent knowledge, hence need to be thoroughly discussed. Similarly educational level emerged as protective factor only for birth outcomes not for maternal complications and cesarean rate needs further discussion. In table four there is repetition, excepting coexistence of two variables rest part has already been covered in table three. In discussion the findings are mostly compared with state statistics. As already commented, the comparison with district figures will be more valid. Reduction in severe anemia may be result of recently introduction of IV iron therapy in PHCs and Rural Hospitals where there is no role of compliance. There may be references stating association between maternal anemia and neonatal mortality. Relation between maternal anemia and neo natal mortality can be explained on birth weight distribution. Mention of increased neonatal mortality among babies born to overweight/obese women may deleted as there was no statistical difference.
--	---

REVIEWER	Dr. Manisha Nair National Perinatal Epidemiology Unit, University of Oxford, UK
REVIEW RETURNED	31-Jan-2018

GENERAL COMMENTS	The paper is interesting and it uses a larger dataset than most other studies in this area. The findings are relevant and helps to understand the problem of anaemia and undernutrition in the study population. I have a few comments and concerns: Methods: The authors mention that BMI was measured at the first antenatal visit, but do not mention at what gestational age. The women are expected to have their first visit during the first trimester (8-12 weeks), but some may register late. Please provide the mean gestational age (with min and max) at which BMI and Hb was measured.
---

	I would think that low BMI is a measure of poor nutrition and anaemia (if all cases are nutritional anaemia) would indicate poor nutrition as well. Combining the two variables could lead to over-adjustment of the model since both variables to a certain degree are measuring nutritional deficiency. The authors report the correlation coefficient for BMI and anaemia as categorical variables. Could you please provide the correlation coefficient for BMI and anaemia as continuous variables? Was a test of interaction conducted to see if anaemia modifies the effect of BMI on the outcomes or vice versa? The results showing the synergistic effect are important, but the justification to generate a combined variable should be based on these tests. Were any other plausible interactions tested? Please specify the name of the test that was used to adjust for clustering. How many hours or minutes after birth, was birthweight measured? Why did the authors not use BMI and Hb concentration as continuous variable? Was there any evidence of the presence of non-linear associations between these risk factors and the outcomes due to which a decision was taken to categorise the data? It would be interesting to examine the risk of adverse outcomes per unit decrease in Hb concentration. Results: Can the authors provide the proportions of moderate and severe anaemia separately in the description of the participants? It would be useful to know the prevalence of severe anaemia. In the footnote of each table, please mention that the analyses accounted for clustering using xxxx method. Figure-1 is confusing. Can this be presented in any other way? Can the authors please show the rates of CS, pregnancy complications, LBW, stillbirth and neonatal death in the study population over time? If these have already been published in a separate paper, please include the details from the paper (citing the paper). Discussion: The prevalence figures (and trends) for anaemia and undernutrition are alarming and important. However, moderate/ severe anaemia is probably more important. Please mention the proportions of moderate/ severe anaemia in addition to the proportion of anaemia (overall) in the discussion section. Please include a detailed limitations section in the discussion. It is important to acknowledge that the study examined a limited number of variables. There are several confounders/ independent risk factors for the outcomes that have not been included in the statistical model, including measures of socioeconomic status, smoking/ tobacco use, ANC visits, etc. There is no evidence that all pregnant women had nutritional anaemia. Are there studies that report prevalence of haemoglobinopathies such as sickle cell anaemia/ disease in the region? Without the evidence of type of anaemia, the recommendations related to iron supplementation programme evaluation is not justified. Other comments: There are some grammatical errors which need to be corrected in the manuscript, particularly in the abstract, and 'Strengths and limitations'. The introduction can be a shorter.
--	--

VERSION 1 – AUTHOR RESPONSE

Reviewer(s)' Comments to Author:

Reviewer: 1

Reviewer Name: Dr. Prakash Prabhakar Rao Doke

Institution and Country: Department of Community Medicine, Bharati Vidyapeeth Deemed University, Medical College, Pune, India

1. Please state any competing interests or state 'None declared': None declared

Response: We have added "None declared" under competing interests.

Please leave your comments for the authors below

Authors are requested to modify as per comments

Reviewer: 2

Reviewer Name: Dr. Manisha Nair

Institution and Country: National Perinatal Epidemiology Unit, University of Oxford, UK

1. Please state any competing interests or state 'None declared': I declare that I have no competing interest

Response: – this has been corrected as described above for Reviewer 1.

Please leave your comments for the authors below

The paper is interesting and it uses a larger dataset than most other studies in this area. The findings are relevant and helps to understand the problem of anaemia and undernutrition in the study population. I have a few comments and concerns:

Methods:

1. The authors mention that BMI was measured at the first antenatal visit, but do not mention at what gestational age. The women are expected to have their first visit during the first trimester (8-12 weeks), but some may register late. Please provide the mean gestational age (with min and max) at which BMI and Hb was measured.

Response: The BMI was measured at the first antenatal care visit when the woman presented for pregnancy registration. Ideally, BMI would have been measured pre-pregnancy or shortly after conception. In our population, although we encourage women in our study catchment area to register as early as possible when they become pregnant, pregnancy registration occurs at varying times during pregnancy from approximately 6 weeks of gestation to close to

term. Specifically, in our study, the range of GA at the first ANC visit was 6 weeks to 38 weeks and the distribution was 6-<20 weeks- 72%; 20-28 weeks - 15%; and >28 weeks - 13%.

We elected to include all pregnant women in this study, regardless of the gestational age during the first ANC visit, as this visit is a defined point to identify women and their fetus at risk of adverse outcomes. We recognized that later in pregnancy, there would be a consistent overestimation of pre-pregnancy BMI, biasing against our study hypothesis that lower BMI is associated with adverse outcomes. However it is a limitation of our study, as we note.

Similarly, maternal hemoglobin was also measured at the first ANC visit rather than pre-pregnancy or shortly after conception. However, maternal hemoglobin is more likely to fall during pregnancy due to plasma volume expansion. We recognize that later in pregnancy, there would be a consistent underestimation of pre-pregnancy anemia, biasing towards our hypothesis that lower hemoglobin is associated with worse maternal and fetal outcomes and this is not only a limitation but also of concern.

Since inclusion of women whose first ANC visit occurred after GA week 20 could result in overestimation of BMI and underestimation of maternal hemoglobin, we analyzed the data both ways - by including all women regardless of which GA they entered the cohort during their first antenatal visit, and, by restricting the cohort to GA between 6 to 20 weeks of pregnancy. Adverse pregnancy outcomes for the entire cohort (all GA) and the cohort of GA 6-20 weeks were similar, indicating that the combination of low hemoglobin and BMI at any point that the woman presents for her first antenatal visit is of clinical and public health importance and requires attention.

For the paper, we elected to present the data for all women regardless of gestational age at presentation so that we can identify the women and fetus at risk during the first ANC visit. We do agree that obtaining pre-pregnancy BMI and maternal hemoglobin or first trimester values would be ideal, but that is very difficult to achieve currently. Future improvements in cost effective non-invasive assessment of hemoglobin in women not using contraception may be useful in the future.

Please see the additions to the text on page 15, lines 3-5.

2. I would think that low BMI is a measure of poor nutrition and anaemia (if all cases are nutritional anaemia) would indicate poor nutrition as well. Combining the two variables could lead to over-adjustment of the model since both variables to a certain degree are measuring nutritional deficiency. The authors report the correlation coefficient for BMI and anaemia as categorical variables. Could you please provide the correlation coefficient for BMI and anaemia as continuous variables? Was a test of interaction conducted to see if anaemia modifies the effect of BMI on the outcomes or vice versa? The results showing the synergistic effect are important, but the justification to generate a combined variable should be based on these tests. Were any other

plausible interactions tested?

Response: We agree that while low BMI and anemia are both markers of poor nutrition, anemia is not only caused by nutritional deficiencies. We have reported the Pearson correlation coefficient for BMI and hemoglobin concentration level as continuous variables in 'Results/Baseline maternal characteristics' section (page 15, lines 7-8). However we preferred to analyze BMI and Anemia as ordinal variables to be consistent with the public health practices and management guidelines that are based on categories of severity of anemia and BMI. The justification to combine binary response BMI and anemia variables was based on the results in Table 3 which indicated an association between BMI and anemia as ordinal variables and birth outcomes. In Table 4 we derived a combination variable for categories of BMI and anemia to provide specific estimates of the effects of the combination of anemia and low BMI on birth outcomes.

Please see the modifications to the text on page 14, lines 9-11.

3. Please specify the name of the test that was used to adjust for clustering.

Response: We used generalized estimating equations to adjust for clustering within the Primary Health Centers. This has been clarified in the 'Methods/Statistical analyses' section.

Please see page 14, lines 6-8.

How many hours or minutes after birth, was birthweight measured?

Response: The birthweight recorded in the registry is measured within 24 hours of birth. This has been added to the 'Methods/Study outcomes' section.

Please see page 13, line 14.

4. Why did the authors not use BMI and Hb concentration as continuous variable? Was there any evidence of the presence of non-linear associations between these risk factors and the outcomes due to which a decision was taken to categorise the data? It would be interesting to examine the risk of adverse outcomes per unit decrease in Hb concentration.

Response: Please see our response to #2 above where we address the analysis of BMI and hemoglobin concentration as ordinal variables to be consistent with existing practices. The associations of BMI and hemoglobin alone and in combinations with the adverse birth outcomes and neonatal death are shown in Tables 3 and 4.

Results:

5. Can the authors provide the proportions of moderate and severe anaemia separately in the description of the participants? It would be useful to know the prevalence of severe anaemia. In the footnote of each table, please mention that the analyses accounted for clustering using xxxx method.

Response: Thank you for this suggestion. We have now separately reported the numbers and frequencies of moderate and severe anemia in the manuscript. This has been added in the "Results/Baseline maternal characteristics" section on page 15, line 8.

As noted above in the response to #3 about clustering, we used generalized estimating equations (GEE) to account for clustering. This has been added to the methods section as well as footnotes of all relevant tables.

6. Figure-1 is confusing. Can this be presented in any other way?

Response: We struggled with the best way to display frequencies of normal, mild and moderate/severe anemia in all 3 categories of BMI by year and have tried other ways to present the data. However, we do believe that this figure effectively shows the trend of anemia stratified by BMI over time.

7. Can the authors please show the rates of CS, pregnancy complications, LBW, stillbirth and neonatal death in the study population over time? If these have already been published in a separate paper, please include the details from the paper (citing the paper).

Response: This is a great point. The rates of all the study outcomes are included in Table 2 in this manuscript and have been discussed in the relevant sections.

Please find the modifications on page 17, lines 9-15.

Discussion:

8. The prevalence figures (and trends) for anaemia and undernutrition are alarming and important. However, moderate/ severe anaemia is probably more important. Please mention the proportions of moderate/ severe anaemia in addition to the proportion of anaemia (overall) in the discussion section.

Please include a detailed limitations section in the discussion. It is important to acknowledge that the study examined a limited number of variables. There are several confounders/ independent

risk factors for the outcomes that have not been included in the statistical model, including measures of socioeconomic status, smoking/ tobacco use, ANC visits, etc.

Response: Thank you for this comment. The prevalence of anemia is presented in the 'Results/Baseline maternal characteristics' section and so we hesitated to add them to the 'Discussion', but for clarity we have added this in section page 22 lines 9-10 .

This was a secondary data analysis of the data collected for the Maternal and Neonatal Health Registry of the US National Institutes of Health "Global Network" as described in the methods section. We acknowledge that there are other confounders of adverse pregnancy outcomes, particularly more detailed information on socioeconomic status, but unfortunately these data were not collected by the Global Network. This limitation has been added to the discussion on page 22, lines 22-23 and page 23, lines 1-2.

9. There is no evidence that all pregnant women had nutritional anaemia. Are there studies that report prevalence of haemoglobinopathies such as sickle cell anaemia/ disease in the region? Without the evidence of type of anaemia, the recommendations related to iron supplementation programme evaluation is not justified.

Response: Thank you for this comment. Although sickle cell anemia, malaria and other parasitic infections can lead to anemia, iron deficiency and undernutrition are responsible for more than 50% of anemia*. Ideally all causes of anemia should be addressed, but given the prevalence of iron deficiency anemia, it is an important public health priority to improve compliance with iron supplementation during pregnancy.

Please see the modification to the text on page 23, lines 15-18.

*Source: Sumarmi S, Puspitarsi N, Handajani N & Wirjatmadi B. Underweight as a Risk Factor for Iron Depletion and Iron-Deficient Erythropoiesis among Young Women in Rural Areas of East Java, Indonesia. *Mal J Nutr* 22(2): 219 - 232, 2016 219

Other comments - from the "Comments BMJ Paper" attachment:

There are some grammatical errors which need to be corrected in the manuscript, particularly in the abstract, and 'Strengths and limitations'.

The introduction can be a shorter.

Response: Thank you for these comments and we apologize for the errors. We have gone through the manuscript very carefully to correct typographical and grammatical errors. We have also rewritten the introduction to make it much shorter as suggested.

1. Title mentions 'rural Maharashtra'; the article pertains to only four districts from Nagpur division.

Response: We have corrected the title to include 'Eastern rural Maharashtra.'

2. Title mentions pregnancy outcomes; however the article does not include abortion which is very important and adverse outcome of pregnancy.

Response:The study focuses on pregnancy outcomes after 20 weeks of gestation. Miscarriages and medical terminations of pregnancy occur before 20 weeks in the gestational period as specified in the 'Methods/Participants' section.

Please see corrections on page 11, lines 4-7.

3. The title says it is a cohort study. In cohort study readers expect analysis of relative risk. The relative risk is mentioned only in abstract and risk ratio at some places in results. Attributable and population attributable risk was expected. The study description does not satisfy the expectations. At the end the check list for STROBE statement for cross sectional studies is given instead of cohort study. There should be some uniformity.

Response:The cohort study STROBE statement has been added and we have carefully checked that we have used the correct terms and made the suggested corrections. The modeling yielded adjusted risk ratios, adjusted for clustering using generalized estimating equations. This is now clearly indicated in the text throughout the manuscript.

4. The objective in abstract mentions 'Central India'. It seems inconsistent.

Response: The objective in the 'Abstract' section has been modified and states 'Eastern Maharashtra'.

5. Mention of mode of delivery as the first main outcome does not get appropriate attention in subsequent writing. If low BMI and anemia are independent variables, mode of delivery may be distantly associated. For mode of delivery availability facilities for cesarean sections is the determining variable.

Response:We have added the details of Cesarean sections to the 'Results/Study Outcomes: Regressions'" section (page 19). The association of BMI and anemia with cesarean sections is also included in Table 3.

6. The result section gives relative risks associated with anemia and underweight separately but risks are not given when they are coexisting. In limitations some mention is there of pre-pregnancy status.

Response: Risks of the combination of anemia and underweight as co-existent conditions are shown in Table 4. This is also now described in the 'Results' section on page 19. Unfortunately, we do not have pre-pregnancy body mass index or hemoglobin levels and have added this to the limitations on page 7, line 8.

7. In introduction SDGs and Global Nutrition Targets are mentioned. Addition of strategies in Indian context will be more relevant. In twelfth plan reduction in anemia in pregnancy is targeted.

Response:The goal of the 12th five-year plan is to attain overall reduction of anaemia in women aged 15-49 years to 28% or half of the current level by 2017. Strategies to do so include making intravenous infusion of iron sucrose for severe anemia available at primary health centers.

This has been added on page 21, lines 15-18.

8. In objectives, the word measure prevalence of anemia may be more suited because reasons, laboratory investigations etc. are not part of study. In second objective the words levels of maternal hemoglobin are used but excepting categorization of anemia no quantitative correlation is attempted. Either the words should be replaced by anemia or depending upon of level hemoglobin serial relative risks may be provided.

Response:Thank you for this suggestion and we have reworded the objectives as recommended. (Page 9, lines 6-9)

9. Study settings may mention named of district. District wise information both from DLHS-4 and NFHS-4 is available. The findings from these reports need to be considered for better comparison.

Response:The study settings now include the names of the 4 districts and we have added district-wise fact sheet figures from NFHS-4 for comparison where they are available.

Please see the modified text on page 9, lines 21-22 and page 22, lines 18-20.

10. The criteria of excluding termination of pregnancy may under estimate the risk of adverse pregnancy outcome associated with anemia/underweight. The women who were followed upto 42 days were eligible for inclusion. Limitations of Sahli's method of estimation of hemoglobin may be have been mentioned. The WHO reference quoted for classification of anemia based up on hemoglobin level is very old may be replaced by latest one. Analysis by three categories of anemia universally accepted may give more meaningful information about graded response.

Response: The inclusion/exclusion numbers from the results section have been updated. Miscarriages and MTPs were excluded from the study for reasons cited above (response to #2). (Page 14, lines 20-22)

Limitations of the Sahli's method have been added in the 'Strengths and limitations section' on page 7 and have been referenced as a source of variation in the hemoglobin levels in 'Discussion' section on page 23, lines 4-5. The standard WHO criteria for classification, though old, remain the same and have been verified from their website and other documentation. These have become widely used and applied in the public health sector and so we have chosen to use them to maintain the universal appeal of our article. Since there are very few women in the severely anemic category, it made more sense to us to combine it with the moderate category and use a two category classification.

11. Secondly the national guidelines for management anemia vary as per severity. The five study outcomes mentioned are not in consonance with objectives. Mode of delivery, pregnancy related complications and neonatal death do not fit aptly under pregnancy outcomes. Instead of classifying mode of delivery into cesarean and non-cesarean, cesarean and vaginal may have been more appropriate terms.

Response: This is a great suggestion. The "mode of delivery" categories are now defined in the 'Methods/Study outcomes' section and we clarified that non-cesarean deliveries include vaginal and assisted vaginal deliveries.

Please see corrections on page 13, line 6-7.

12. In result section one of the reasons given for exclusion is no pregnancy outcome yet. This is contradictory to inclusion criteria. These women did not meet criteria of inclusion. Probably these were the pregnant women enrolled in last few months. They need not be included in first instance.

Response: We thank the reviewer for this important point and have modified the results section and fixed the description of the included women to address incomplete follow-up of the women at that time of analysis. (page 14, lines 20-22)

13. A slight but significant correlation was observed which may be due to large number. The observation of more probability of maternal complications and cesarean rate among non-anemic women and less in anemic is contradictory to prevalent knowledge, hence need to be thoroughly discussed.

Response:Non-anemic women are also those with a higher BMI who are at an increased risk of miscarriage, gestational diabetes, preeclampsia, venous thromboembolism, induced labour, caesarean section, anaesthetic complications and wound infections, and they are less likely to initiate or maintain breastfeeding. Babies of obese mothers are at increased risk of stillbirth, congenital anomalies, prematurity, macrosomia and neonatal death.

We have added this on page 24, lines 14-19.

14. Similarly educational level emerged as protective factor only for birth outcomes not for maternal complications and cesarean rate needs further discussion.

Response:Regardless of educational levels, there is no restriction of rural women from equitable access and incentives to deliver in health facilities. Women also receive access to cesarean facilities through the National Rural Health Mission. Therefore, education no longer remains a predictor and may not have an impact on pregnancy related complications and cesarean section rates.

15. In table four there is repetition, excepting coexistence of two variables rest part has already been covered in table three. In discussion the findings are mostly compared with state statistics. As already commented, the comparison with district figures will be more valid.

Response:Table 4 mentions the same baseline characteristics used in Table 3 to show changes in the risks for these characteristics if the BMI and anemia variables are modified to an interaction variable. Table 4 shows that there are no prominent changes in the risks based on other baseline characteristics, even if the BMI/anemia combination variable is used. The district figures are already updated in the manuscript (please see above response to # 9).

16. Reduction in severe anemia may be result of recently introduction of IV iron therapy in PHCs and Rural Hospitals where there is no role of compliance. There may be references stating association between maternal anemia and neonatal mortality. Relation between maternal anemia and neo natal mortality can be explained on birth weight distribution.

Response: We agree with this comment but do not yet have any data to address this point. We have not found any references that address this either. It is an important future direction.

17. Mention of increased neonatal mortality among babies born to overweight/obese women may be deleted as there was no statistical difference.

Response: We agree – thank you - and have removed from the article as suggested.

VERSION 2 – REVIEW

REVIEWER	Dr. Prakah Prabhakarrao Doke Community Medicine Department, Bharati Vidyapeeth Deemed University Medical College, Pune, India
REVIEW RETURNED	14-Mar-2018

GENERAL COMMENTS	I did not find convincing argument to continue last three columns in table three. Because they are almost same to table four. The heading, the figures and the analysis is similar. My suggestion is that Table four may be reduced to last five rows explaining combination of anemia and BMI.
---

REVIEWER	Manisha Nair University of Oxford, UK
REVIEW RETURNED	16-Mar-2018

GENERAL COMMENTS	I thank the authors for their response and corrections. I still have a few concerns and would suggest that these be addressed before the paper is accepted for publication.  1. The authors have included information about gestational age at first visit in the revised draft. Although the authors acknowledge the limitations of the broad range of GA during first visit in their response, this is not included in the limitations section in the paper. The broad range of GA during first visit is a major problem since the pathophysiology underlying the association between anaemia and maternal and fetal outcomes, and between low BMI and outcomes at a late GA is much different from the pathophysiology at early gestation. BMI measurements late in pregnancy cannot be a reliable measure of nutrition, but more likely to be due to fetal growth faltering/ low liquor volume, etc which are known risk factors for stillbirth and perinatal mortality, rather than BMI. Inclusion of BMI as a main exposure variable looks problematic. Instead of combining the data by stating that results from separate models for 6-20 weeks and late GA were not different, it might be best to present them as separate models (although still not ideal, as the authors mention in their response). This will at least provide an opportunity in the discussion section to explain the plausible underlying pathophysiology for the associations at early and late GA. 2. With regard to combining the variables anaemia and BMI, the authors mention that this was based on the results shown in Table-3 – an association between anaemia and BMI. I am sorry, but I cannot find the results in Table-3. The recalculated correlation coefficient using Hb and BMI as continuous variables shows that the correlation is not strong between the two variables ($r=0.2$). The correlation is significant ($p<0.001$), but it is not a strong correlation. It is important that the authors test for interaction between anaemia and BMI, as well as report other plausible interactions. A significant interaction between the anaemia and BMI variables will justify the combined variable. Again, I would suggest that the analysis be done on two sets of data – pregnant women with BMI 6-20 weeks and late GA. 3. Table-3 – is the association between stillbirth and underweight not significant? The 95% CI touches 1.00, but does not cross. 4. With regard to my comment on authors' strong inference on lack of compliance to iron supplementation as the main reason for high prevalence of anaemia and therefore a strong recommendation for
--

	iron supplementation – the response is acceptable. I would suggest that the authors copy the text from their response into the paper. 5. There are two points from Reviewer-1 which the authors may want to reconsider – a. The reviewer is right in anticipating that the association between anaemia and C-section is highly likely to be biased by the availability of facilities for C-section. b. Regarding the WHO classification for anaemia – although this is still accepted for categorising anaemia, an effect of Hb as a continuous variable on maternal and fetal outcomes is more meaningful. This also improves the statistical power of the analysis. Can the authors add the mean and inter-quartile range for Hb concentration and BMI?
--	---

VERSION 2 – AUTHOR RESPONSE

Reviewer(s)' Comments to Author:

Reviewer: 1

Reviewer Name: Dr.PrakahPrabhakarraoDoke

Institution and Country: Community Medicine Department, BharatiVidyapeeth Deemed University Medical College, Pune, India

Please state any competing interests or state 'None declared': None declared

Please leave your comments for the authors below

1. I did not find convincing argument to continue last three columns in table three. Because they are almost same to table four. The heading, the figures and the analysis is similar. My suggestion is that Table four may be reduced to last five rows explaining combination of anaemia and BMI.

Response: We would like to thank the reviewer for his comments. Table 4 reports the impact of anaemia vs. no anemia, underweight vs. underweight and a combination of anemia and underweight on 3 outcomes - stillbirths, neonatal deaths and low birth weight. The data in Table 4 are relevant from the public health point of view and are different from the results presented in Table 3. Table 3 shows the risk of anemia (not anemic vs. mild anemia vs. moderate/severe anemia) and BMI (underweight vs. normal weight vs. overweight/obese), separately (not combined) on 5 outcomes – caesarean section, pregnancy related maternal complications, stillbirths, neonatal deaths and low birth weight.

Reviewer: 2

Reviewer Name: Manisha Nair

Institution and Country: University of Oxford, UK

Please state any competing interests or state 'None declared': None declared

Overall Response to Reviewer 2:

We thank Reviewer 2 for her comments to substantially improve the analysis and the manuscript -

Based on the comments and suggestions, we have conducted additional analysis. They are as follows –

1. Table 1 - we have now added a 4th row to report the proportion of pregnant women who had either anemia or underweight alone or in combination, across the 8 years. This indicates the burden of this health problem in the study population, from a public health perspective.
2. As requested, we have added the mean HB, Mean BMI and their IQR in Table 1.
3. We have now added additional row for each outcome in Table 2. This additional row shows the rates of the outcome when a pregnant woman is both anemic and underweight.
4. We examined the interaction of Hemoglobin X BMI, as continuous variables for all the five outcomes. The results are as follows –

Table: Continuous hemoglobin and BMI and their interactions as Risk factors for study outcomes (All included women)

Characteristics	Study Outcomes				
	Cesarean Sections	Pregnancy related Maternal complications	Stillbirths	Neonatal deaths (within 28 days of birth)	Low birth weight (<2500 gms)
	N= 4816, 20%	N=3260, 14%	N=488, 2% RR(95% CI)^	N=390, 1% RR(95% CI)^	N=3776, 16% RR(95% CI)^
	RR(95% CI)^	RR(95% CI)^			
Hemoglobin level (gm/dl)	1(0.94 - 1.05)	1.02(0.95 - 1.09)	1.02(0.83 - 1.24)	1.12(0.91 - 1.39)	1.00(0.93 - 1.08)
Body Mass Index (kg/m2)	0.97(0.87 - 1.09)	0.97(0.84 - 1.12)	0.91(0.61 - 1.35)	1.02(0.66 - 1.57)	0.98(0.84 - 1.13)
c.bmi#c.hb	1.01*(1.00 - 1.01)	1(1.00 - 1.01)	1(0.98 - 1.02)	0.99(0.97 - 1.01)	1.00(0.99 - 1.00)

* P<0.05, ** P<0.001

^ Obtained by multivariable GEE regression models for each outcome, separately

5. To improve the analysis of the impact of anemia and BMI on the 5 study outcomes, we divided the data into two data sets according to the GA of pregnant women at enrolment – ≤ 20 weeks and > 20 weeks of GA. The results are now reported in the manuscript on page no 22-24 as table 3A, 3B and 3C.

Please leave your comments for the authors below

I thank the authors for their response and corrections. I still have a few concerns and would suggest that these be addressed before the paper is accepted for publication.

1. The authors have included information about gestational age at first visit in the revised draft. Although the authors acknowledge the limitations of the broad range of GA during first visit in their response, this is not included in the limitations section in the paper. The broad range of GA during first visit is a major problem since the pathophysiology underlying the association between anaemia and maternal and fetal outcomes, and between low BMI and outcomes at a late GA is much different from the pathophysiology at early gestation. BMI measurements late in pregnancy cannot be a reliable measure of nutrition, but more likely to be due to fetal growth faltering/ low liquor volume, etc which are known risk factors for stillbirth and perinatal mortality, rather than BMI. Inclusion of BMI as a main exposure variable looks problematic. Instead of combining the data by stating that results from separate models for 6-20 weeks and late GA were not different, it might be best to present them as separate models (although still not ideal, as the authors mention in their response). This will at least provide an opportunity in the discussion section to explain the plausible underlying pathophysiology for the associations at early and late GA.

Response: We thank the reviewer for this comment and we agree with the reviewer that broad range of GA during first visit has an effect on the maternal and foetal outcomes. We have also analysed the data according to gestational age categories of ≤ 20 weeks and > 20 weeks of gestational age and have stated the factors affecting the main outcomes accordingly. Please see the changes on page no 21 for the description and page no 22-24 for tables (3A, 3B and 3C). We also added the broad range of gestational age to the limitations on page no 7, line no 9-11.

We elected to keep the ordinal variables for anemia and BMI as they are important from public health viewpoint. However, we have added continuous BMI and HB along with their combination variable in Table 1 on page 17 in the manuscript. Additionally, we also have included regression result tables when haemoglobin and BMI are analysed as continuous variables at the end of this document. The results are similar regardless of whether haemoglobin and BMI are treated as continuous or ordinal variables.

2. With regard to combining the variables anaemia and BMI, the authors mention that this was based on the results shown in Table-3 – an association between anaemia and BMI. I am sorry, but I cannot find the results in Table-3. The recalculated correlation coefficient using Hb and BMI as continuous variables shows that the correlation is not strong between the two variables ($r=0.2$). The correlation is significant ($p<0.001$), but it is not a strong correlation. It is important that the authors test for interaction between anaemia and BMI, as well as report other plausible interactions. A significant interaction between the anaemia and BMI variables will justify the combined variable. Again, I would suggest that the analysis be done on two sets of data – pregnant women with BMI 6-20 weeks and late GA.

Response: We agree with the reviewer about the interpretation of the correlation between BMI and anemia as continuous variables. However, we have already presented that there is a significant association between BMI and anemia by using a cluster adjusted multinomial regression model described in the manuscript on page 16, lines 6-10. The results presented there show strong associations between ordinal categories of BMI and hb from Table1. This implies that the association between BMI and anemia, should remain significant even if the ordinal categories are collapsed to

binomial. This should justify the BMI-anemia combination variable without the need for additional testing.

3. Table-3 – is the association between stillbirth and underweight not significant? The 95% CI touches 1.00, but does not cross.

Response: We revised the analysis and stratified by gestational age of ≤ 20 weeks and > 20 weeks and found that stillbirths were not associated with underweight recorded at any time during pregnancy, nor by mild anemia when enrolled < 20 weeks. This is seen in Tables 3A and 3B on pages 22, 23.

4. With regard to my comment on authors' strong inference on lack of compliance to iron supplementation as the main reason for high prevalence of anaemia and therefore a strong recommendation for iron supplementation – the response is acceptable. I would suggest that the authors copy the text from their response into the paper.

Response: We thank the reviewer for the comment. We have now added this to the paper on page no 26; line no 14-18 and on page no 27 on lines no 16-21

5. There are two points from Reviewer-1 which the authors may want to reconsider –

a. The reviewer is right in anticipating that the association between anaemia and C-section is highly likely to be biased by the availability of facilities for C-section.

Response: The rates of C-section in our study area have increased over the years due to increased availability of facilities able to perform C-sections. However tables 3A and 3B indicates that women who were anemic at any time during pregnancy (≤ 20 weeks and > 20 weeks) were less likely to undergo C-sections. Please see the additional text on page no 28, lines 15-18.

b. Regarding the WHO classification for anaemia – although this is still accepted for categorising anaemia, an effect of Hb as a continuous variable on maternal and fetal outcomes is more meaningful. This also improves the statistical power of the analysis. Can the authors add the mean and inter-quartile range for Hb concentration and BMI?

Response: The means, standard deviation and the inter quartile range for Hb concentration and BMI have now been added to the Table 1 of baseline characteristics on page no 17. We have analyzed the effects of BMI and HB as continuous variables on maternal and fetal outcomes and have included the results in this document, as well.

Table 3A: Risk factors for study outcomes, multivariable estimates (GA at enrolment ≤ 20 weeks)

Characteristics	Study Outcomes				
	Cesarean Sections	Pregnancy related Maternal complications	Stillbirths	Neonatal deaths (within 28 days of birth)	Low birth weight (<2500 gms)
	N=11628, 24% RR(95% CI)^	N=6762, 14% RR(95% CI)^	N=1104, 2.3% RR(95% CI)^	N=963, 2.0% RR(95% CI)^	N=8378, 18% RR(95% CI)^
Mother's Age (years)					
<20	0.83*(0.73 - 0.95)	0.84*(0.72 - 0.97)	0.73(0.47 - 1.13)	1.23(0.87 - 1.75)	1(0.88 - 1.15)
20-29	ref	ref	ref	ref	ref
>29	1.52** (1.41 - 1.64)	1.34** (1.20 - 1.49)	1.44* (1.13 - 1.85)	1.35* (1.02 - 1.79)	1.29** (1.16 - 1.42)
Mother's Education (grades)					
Primary or Less (<=4)	0.50** (0.47 - 0.54)	0.76** (0.70 - 0.83)	1.65** (1.36 - 2.01)	1.49** (1.22 - 1.83)	1.15** (1.07 - 1.23)
Secondary(5-10)	0.70** (0.67 - 0.73)	0.90** (0.85 - 0.94)	1.39** (1.20 - 1.60)	1.29** (1.11 - 1.51)	1.14** (1.09 - 1.20)
University(>10)	ref	ref	ref	ref	ref
Parity					
Nulliparous	1.34** (1.29 - 1.39)	2.01** (1.87 - 2.16)	1.32** (1.17 - 1.50)	1.57** (1.38 - 1.80)	1.37** (1.31 - 1.44)
1-2	ref	ref	ref	ref	ref
>2	0.49** (0.40 - 0.60)	0.80* (0.64 - 1.00)	1.49* (1.05 - 2.11)	1.60* (1.09 - 2.35)	0.99 (0.84 - 1.16)
Hemoglobin level (gm/dl)	1.07** (1.06 - 1.06)	1.06** (1.05 - 1.07)	1 (0.98 - 1.03)	1.01 (0.99 - 1.03)	0.96** (0.95 - 0.97)

	1.07)				
Body Mass Index (kg/m ²)	1.04** (1.02 - 1.06)	1(0.97 - 1.03)	0.93(0.86 - 1.00)	0.85** (0.79 - 0.92)	0.91** (0.88 - 0.93)

* P<0.05, ** P<0.001

^ Obtained by multivariable GEE regression models for each outcome, separately

Table B: Risk factors for study outcomes, multivariable estimates (GA at enrolment > 20 weeks)

Characteristics	Study Outcomes				
	Cesarean Sections N=4816, 20% RR(95% CI)^	Pregnancy related Maternal complications N=3260, 14% RR(95% CI)^	Stillbirths N=488, 2% RR(95% CI)^	Neonatal deaths (within 28 days of birth) N=390, 1% RR(95% CI)^	Low birth weight (<2500 gms) N=3776, 16% RR(95% CI)^
Mother's Age (years)					
<20	0.65*(0.47 - 0.88)	0.8(0.59 - 1.09)	1.25(0.64 - 2.42)	1.43(0.73 - 2.81)	1.15(0.91 - 1.46)
20-29	ref	ref	ref	ref	ref
>29	1.62** (1.45 - 1.82)	1.35** (1.16 - 1.57)	2.31** (1.70 - 3.13)	1.49* (1.01 - 2.19)	1.32** (1.15 - 1.51)
Mother's Education (grades)					
Primary or Less (<=4)	0.45** (0.41 - 0.50)	0.61** (0.54 - 0.68)	1.47* (1.11 - 1.96)	1.49* (1.08 - 2.04)	1.05 (0.95 - 1.16)
Secondary(5-10)	0.67** (0.63 - 0.71)	0.87** (0.80 - 0.94)	1.32* (1.04 - 1.69)	1.34* (1.02 - 1.76)	1.08 (1.00 - 1.17)

University(>10)	ref	ref	ref	ref	ref
Parity					
Nulliparous	1.50**(1.41 - 1.60)	1.99**(1.83 - 2.16)	1.25*(1.04 - 1.51)	1.30*(1.06 - 1.60)	1.31**(1.23 - 1.40)
1-2	ref	ref	ref	ref	ref
>2	0.42**(0.32 - 0.56)	0.75*(0.57 - 0.98)	0.87(0.54 - 1.38)	1.44(0.92 - 2.27)	0.98(0.82 - 1.18)
Hemoglobin level (gm/dl)	1.07**(1.06 - 1.08)	1.07**(1.06 - 1.08)	0.95*(0.92 - 0.99)	0.96(0.93 - 1.00)	0.94**(0.93 - 0.96)
Body Mass Index (kg/m ²)	1.08**(1.04 - 1.12)	1.01(0.97 - 1.05)	0.75**(0.68 - 0.83)	0.68**(0.61 - 0.77)	0.87**(0.83 - 0.90)

* P<0.05, ** P<0.001

^ Obtained by multivariable GEE regression models for each outcome, separately

Table C: Risk factors for study outcomes, multivariable estimates (all included)

Characteristics	Study Outcomes				
	Cesarean Sections	Pregnancy related Maternal complications	Stillbirths	Neonatal deaths (within 28 days of birth)	Low birth weight (<2500 gms)
	N=16693, 23% RR(95% CI)^	N=10163, 14% RR(95% CI)^	N=1620, 2.2% RR(95% CI)^	N=1368, 1.9% RR(95% CI)^	N=12347, 17% RR(95% CI)^
Mother's Age (years)					
<20	0.81**(0.72 - 0.91)	0.84*(0.73 - 0.95)	0.88(0.62 - 1.25)	1.28(0.94 - 1.75)	1.03(0.92 - 1.16)
20-29	ref	ref	ref	ref	ref

>29	1.57**(1.47 - 1.67)	1.36**(1.24 - 1.48)	1.76**(1.45 - 2.12)	1.47**(1.18 - 1.84)	1.30**(1.20 - 1.41)
Mother's Education (grades)					
Primary or Less (≤4)	0.47**(0.44 - 0.50)	0.69**(0.64 - 0.74)	1.59**(1.36 - 1.86)	1.45**(1.22 - 1.72)	1.12**(1.06 - 1.19)
Secondary(5-10)	0.68**(0.66 - 0.71)	0.89**(0.85 - 0.93)	1.37**(1.21 - 1.55)	1.29**(1.13 - 1.47)	1.13**(1.08 - 1.17)
University(>10)	ref	ref	ref	ref	ref
Parity					
Nulliparous	1.39**(1.34 - 1.44)	2.01**(1.89 - 2.13)	1.31**(1.18 - 1.45)	1.48**(1.32 - 1.66)	1.35**(1.30 - 1.41)
1-2	ref	ref	ref	ref	ref
>2	0.45**(0.38 - 0.53)	0.76*(0.64 - 0.90)	1.21(0.91 - 1.59)	1.50*(1.12 - 2.00)	0.99(0.88 - 1.12)
Anemia (gm/dl)	1.06**(1.05 - 1.06)	1.06**(1.05 - 1.06)	0.98(0.97 - 1.00)	0.99(0.97 - 1.01)	0.96**(0.95 - 0.96)
Body Mass Index (kg/m ²)	1.05**(1.03 - 1.07)	1(0.98 - 1.02)	0.87***(0.82 - 0.92)	0.80***(0.75 - 0.85)	0.90***(0.88 - 0.92)

* P<0.05, ** P<0.001

^ Obtained by multivariable GEE regression models for each outcome, separately

VERSION 3 – REVIEW

REVIEWER	Manisha Nair University of Oxford, UK
REVIEW RETURNED	25-May-2018

GENERAL COMMENTS	I thank the authors for revising the paper as suggested. Clearly, you can now see the differential effect of Hb and BMI based on GA. This needs highlighting referring to any underlying patho-physiology. A few more minor comments. 1. In the statistical analysis section (page 14, lines 7-8) the authors state that they calculated the correlation coefficient for hemoglobin concentrations and BMI, but I cannot find the results. Regardless of
---

	any other test, it might be good to add the result to provide more information to the readers. 2. Please add the results of non parametric test for trend over years of enrollment in Table-2 for the outcomes as is done in Table-1. 3. In the section 'study outcomes: Regression', please check that you are reporting risk of an outcome and not rate - page 22, line 10 4. Page 29, lines 3-4 - what do the authors mean by 'one public health action'? 5. Page 30, line 10 what do authors mean by 'forgetfulness'? 6. Page 34, line 1 - please change to 'The risk of neonatal death...' 7. Page 34, line 5 - please change to 'odds of neonatal mortality to be 2.7 times higher among...' I would request the editor to accept the manuscript once these minor changes have been made.
--	---

VERSION 3 – AUTHOR RESPONSE

Reviewer: 2

Reviewer Name: Manisha Nair

Institution and Country: University of Oxford, UK

Please state any competing interests or state 'None declared': None declared.

Please leave your comments for the authors below

I thank the authors for revising the paper as suggested. Clearly, you can now see the differential effect of Hb and BMI based on GA. This needs highlighting referring to any underlying pathophysiology.

Response: The underlying pathophysiology for the differential effect of Hb and BMI based on GA has been described on page no 26, line no 17-21 and on page no 28, line no 11-15. The specific text added on page no 26 is “Though anemia occurring anytime during pregnancy is a risk factor for poor neonatal outcomes, anemia especially during the third trimester is an important factor in determining birth weight. Rapid fetal growth occurs in the third trimester, increasing the iron and other micronutrient requirement. This pathophysiology explains the association of third trimester hemoglobin levels with low birth weight and neonatal deaths.” The specific text added on page 28 is “Maternal pre-pregnancy BMI as well as BMI during any trimester of pregnancy has different effects on maternal and fetal outcomes. Adverse effects of low BMI preferentially target the fetus, by causing growth restriction and LBW, whereas high BMI has a greater impact on maternal health and subsequently affects neonatal health through altered glucose homeostasis, leading to fetal macrosomia and other such pathophysiological mechanisms. [44-46]

A few more minor comments.

1. In the statistical analysis section (page 14, lines 7-8) the authors state that they calculated the correlation coefficient for hemoglobin concentrations and BMI, but I cannot find the results. Regardless of any other test, it might be good to add the result to provide more information to the readers.

Response: The correlation coefficient and its significance test results have already been stated in the "Results/Baseline maternal characteristics" section on page 16 - lines 5,6.

2. Please add the results of non parametric test for trend over years of enrollment in Table-2 for the outcomes as is done in Table-1.

Response: Results for Cuzick's non parametric test for trend over years of enrollment have now been added to Table 2.

3. In the section 'study outcomes: Regression', please check that you are reporting risk of an outcome and not rate - page 22, line 10

Response:Rate has been corrected to risk as we agree that we are reporting risk of outcome.

4. Page 29, lines 3-4 - what do the authors mean by 'one public health action'?

Response: We have modified these words in accordance with the comment.

5. Page 30, line 10 what do authors mean by 'forgetfulness'?

Response:We have clarified forgetfulness and added after forgetfulness "to take timely dosage of iron supplementation pills".

6. Page 34, line 1 - please change to 'The risk of neonatal death...'

Response: We have made this correction.

7. Page 34, line 5 - please change to 'odds of neonatal mortality to be 2.7 times higher among...'

Response: We have made this correction.